# Global disparities in SARS-CoV-2 genomic surveillance

Genomic sequencing is essential to track the evolution and spread of SARS-CoV-2, optimize molecular tests, treatments, vaccines, and guide public health responses. To investigate the global SARS-CoV-2 genomic surveillance, we used sequences shared via GISAID to estimate the impact of sequencing intensity and turnaround times on variant detection in 189 countries. In the first two years of the pandemic, 78% of high-income countries sequenced >0.5% of their COVID-19 cases, while 42% of low- and middle-income countries reached that mark. Around 25% of the genomes from high income countries were submitted within 21 days, a pattern observed in 5% of the genomes from low- and middle-income countries. We found that sequencing around 0.5% of the cases, with a turnaround time <21 days, could provide a benchmark for SARS-CoV-2 genomic surveillance. Socioeconomic inequalities undermine the global pandemic preparedness, and efforts must be made to support low- and middle-income countries improve their local sequencing capacity.

More than 2 years into the COVID-19 pandemic, many countries continue to face large epidemics of SARS-CoV-2 infections[1], mostly driven by the emergence and spread of novel viral variants[2], and unequal access to vaccines, especially earlier in the pandemic[3–6]. Genomic surveillance has been critical to study many rapidly evolving pathogens[7], and has been employed to investigate SARS-CoV-2 evolution and spread, to design and optimize diagnostic tools and vaccines, and to rapidly identify and assess viral lineages with altered epidemiological characteristics, including variants of concern (VOCs) such as Alpha/B.1.1.7, Beta/B.1.351, Gamma/P.1, Delta/B.1.617.2 and Omicron/B.1.1.529. These lineages pose increased global public health risks due to their greater transmissibility and potential immune escape from neutralizing antibodies induced by natural infections and/or vaccines[8,9]. Variants of interest (VOIs) also require continued monitoring for changes in transmissibility, disease severity, or antigenicity[10]. Such variants with higher epidemic potential have been demanding more specific measures, proportional to the risk posed by them, and to do so, policy makers need to know "what" pathogen is present locally, "where" it circulates in the community, "when" such variants may arrive, "why" they represent more risks, and "who" is most at risk[11]. Without answers to these questions, efficient public health policies cannot be implemented, and lives are unnecessarily impacted (high morbidity: long COVID,

sequelaes) or lost (high mortality). Throughout this pandemic, genomic information has been instrumental for planning measures to curb the impacts of variants in low-, middle- and high income countries that implemented evidence-based policies in response to the emergence and spread of VOCs[12–27]. To help guide public health responses to evolving variants, it is essential to track the diversity of SARS-CoV-2 lineages circulating worldwide in near real-time[8,28,29]. Data generators around the world have been submitting an unprecedented number of SARS-CoV-2 genomes in publicly-accessible databases: up to June 9th, 2022, >11.3 million consensus sequences (FASTA) were shared via the EpiCoV database hosted by the GISAID Data Science Initiative[30]. Over 5.5 million sequences can also be found in the archives of the International Nucleotide Sequence Database Collaboration[31] together with >4.5 million raw read sequences (FASTQ)[32]. By way of comparison, 1,614,498 influenza sequences have been shared via GISAID since 2008[33]. Despite improvements in models for equitable sharing of pathogen genomic data[34], there are striking differences in the intensity of genomic surveillance worldwide. Here we examine global publicly-accessible SARS-CoV-2 genomic surveillance data 2 years of COVID-19 pandemic (from March 2020 to February 2022) to identify key aspects associated with sequencing intensity and timely variant detection, and investigate the consequences of surveillance disparities.

✉ e-mail: andersonfbrito@gmail.com; nfaria@ic.ac.uk

# Results

## Global disparities in SARS-CoV-2 genomic surveillance

To investigate spatial and temporal heterogeneity in SARS-CoV-2 genome sequencing intensity, we explored the percentage of COVID-19 cases sequenced each week per country from March 2020 to February 2022 (Fig. 1 and Supplementary Data 1). It has been proposed that at least 5% of SARS-CoV-2 positive samples should be sequenced to detect viral lineages at a prevalence of 0.1 to 1.0%[35], but we identified that only 13 out of 189 countries (6.8%) worldwide had 5% or more of their total confirmed cases sequenced, while 86 out of 189 countries

had <0.5% of confirmed cases sequenced (Figs. 1 and 2A and S1). Throughout the first 2 years of pandemic, only seven countries or territories depended mostly on the sequencing capacity from other countries, having 25% or more of their genomes sequenced abroad (Fig. S2 and Supplementary Data 2). Until late February 2022, while the total number of reported cases was relatively similar in high-income countries (HICs) and low/middle-income countries (LMICs) (i.e., 232.7 and 199.1 million cases, respectively), HICs submitted 10-fold more sequences per COVID-19 case (3.53% and 0.35% sequenced cases, respectively) (Supplementary Data 3). Countries that faced mostly

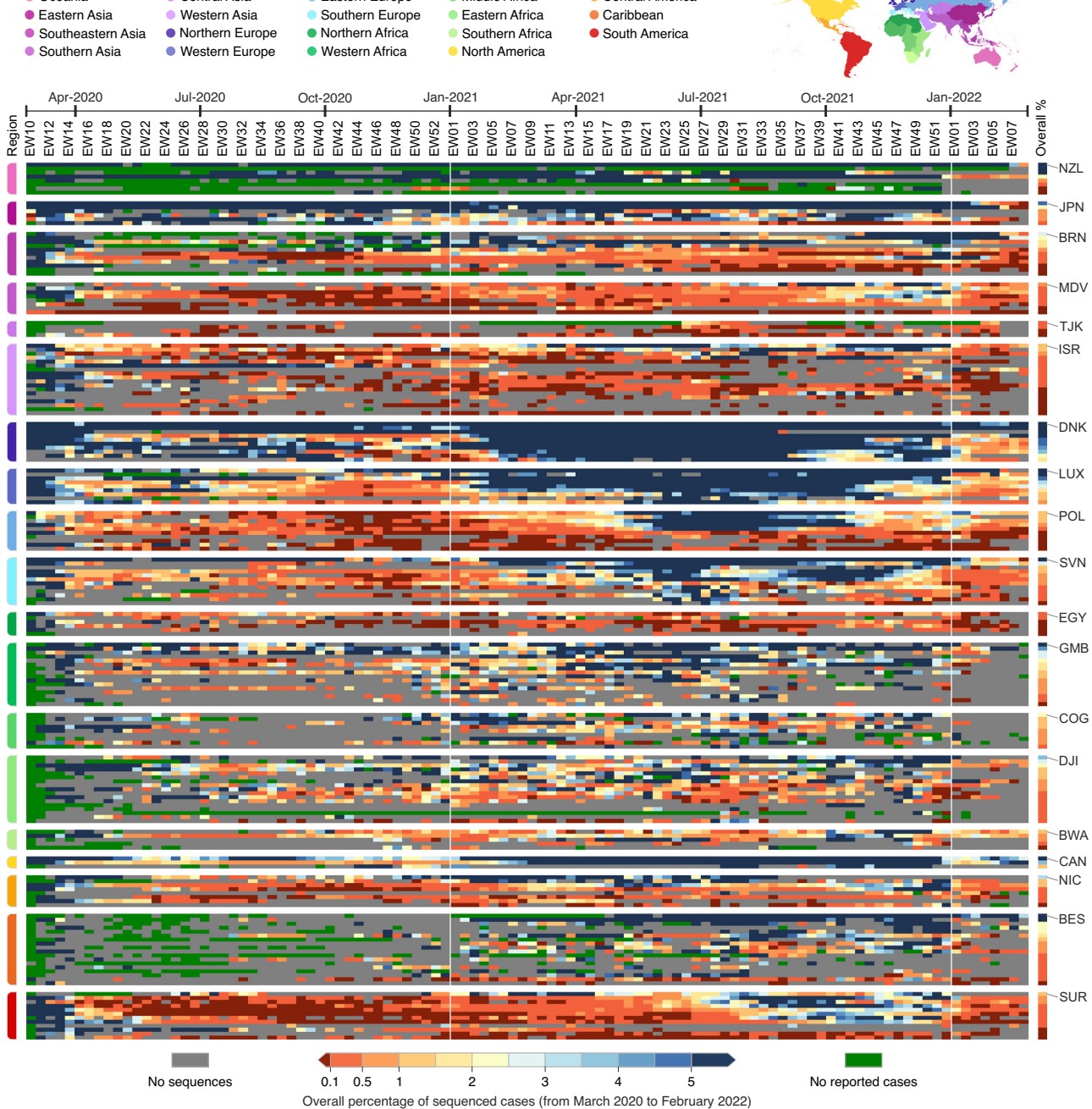

**Fig. 1 | Disparities in SARS-CoV-2 global genomic surveillance.** Percentage of reported cases that were sequenced per country, per epidemiological week (EW), based on genomes collected from EW 10 of 2020 (March 1st) to EW 8 of 2022 (February 26th), with metadata submitted to GISAID up to March 18th, 2022. Updated numbers on sequence submissions and proportion of sequenced cases are available on the GISAID Submissions Dashboard at "gisaid.org". Countries are grouped in regions according to the UNSD geoscheme, and countries with the

highest overall proportion of sequenced cases are highlighted using the ISO 3166-1 nomenclature: NZL New Zealand, JPN Japan, BRN Brunei, MDV Maldives, TJK Tajikistan, ISR Israel, DNK Denmark, LUX Luxembourg, POL Poland, SVN Slovenia, EGY Egypt, GMB Gambia, COG Republic of the Congo, DJI Djibouti, BWA Botswana, CAN Canada, NIC Nicaragua, BES Bonaire, and SUR Suriname.

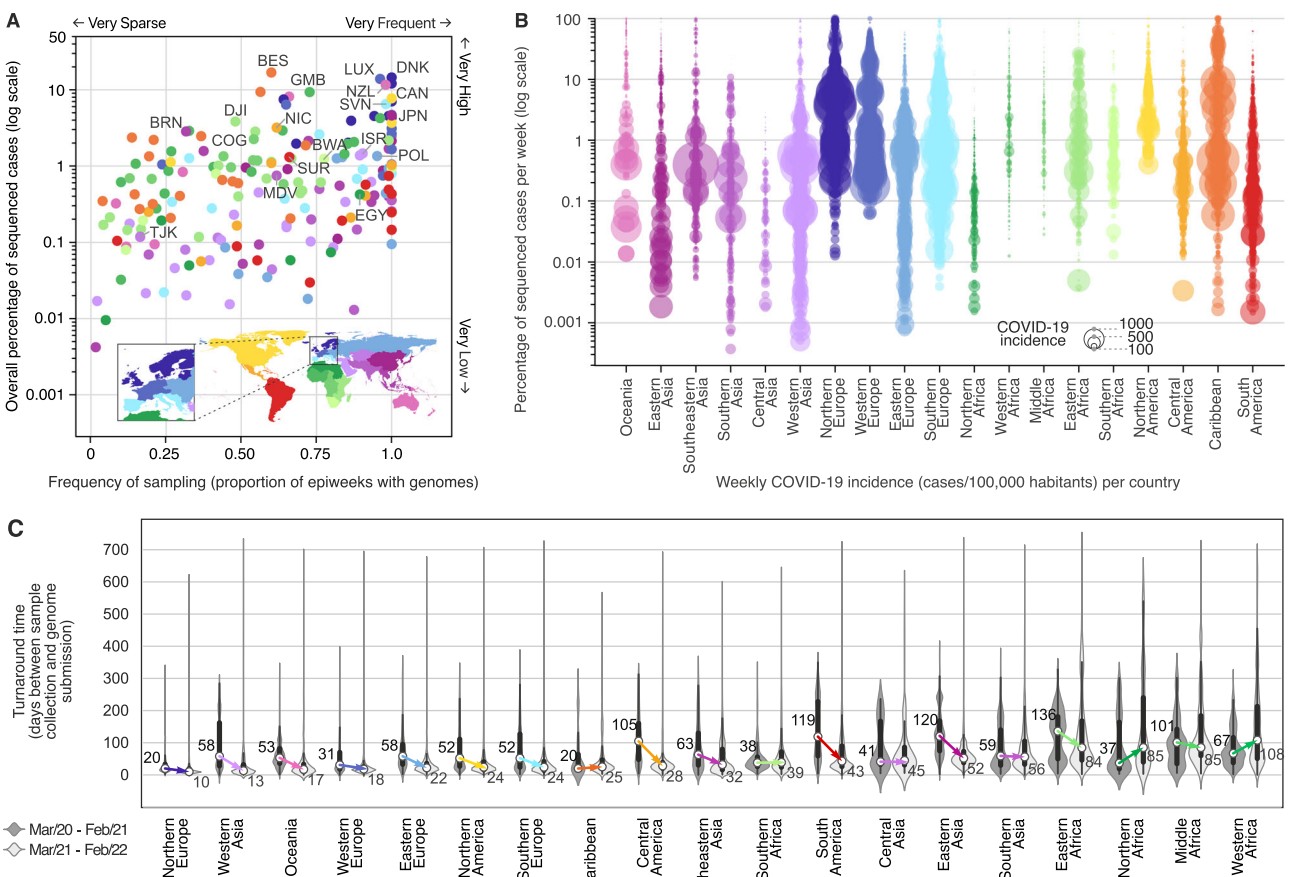

**Fig. 2 | Genomic sequencing intensity and timeliness. A** Frequency and overall percentage of sequenced cases per country (colored as in Fig. 1). This plot summarizes the data shown in Fig. 1, where the *x*-axis shows the percentage of EWs with sequenced cases, and the *y*-axis displays the overall percentage of cases (shown in Fig. 1 as the rightmost column). Countries with the highest overall percentage of sequenced cases in each region are highlighted using the ISO 3166-1 nomenclature: NZL New Zealand, JPN Japan, BRN Brunei, MDV Maldives, TJK Tajikistan, ISR Israel, DNK Denmark, LUX Luxembourg, POL Poland, SVN Slovenia, EGY Egypt, GMB Gambia, COG Republic of the Congo, DJI Djibuti, BWA Botswana, CAN Canada, NIC Nicaragua, BES Bonaire, and SUR Suriname. **B** Percentage of cases sequenced per EW per country, per geographic region. Each circle represents an EW with at least one sequenced case; circle diameters represent incidence, defined here as number of reported cases per 100,000 people per EW per country. **C** Distribution of turnaround times of genomes collected in different geographic regions during the first year (from March 2020 to February 2021) and second year (from March 2021 to February 2022) of COVID-19 pandemic, grouped by year of submission (*n* = 8,947,455 genomes). The elements in the violin plots represent the median TATs (white circles), the interquartile range (black rectangles) and the minimum and maximum data points in the datasets (black vertical lines). The arrows highlight the changes in the median TATs between the first and second year of pandemic.

moderate or lower incidences (<100 cases per 100,000 people) were able to sequence higher proportions of cases (Figs. 1B and S3 and S4). Exceptionally, some countries, such as Denmark, Japan and the UK, despite facing scenarios of high weekly COVID-19 incidence (>100 cases per 100,000 people) in the first 2 years of the pandemic, were still able to maintain sequencing intensity >5% in most weeks (Figs. 1 and 2A, B and S4).

Many countries in Africa and Asia, despite reporting low COVID-19 incidences in most weeks (<10 weekly cases per 100,000 people, see Figs. S3 and S4), have not reached levels of genomic surveillance similar to Japan (4.6%), Gambia (9.3%), or New Zealand (11.6%), which experienced similarly low COVID-19 incidences during the first 2 years of pandemic (Figs. 2A, B and S3 and S4). As we show in the next sections, socioeconomic factors may explain these disparities among countries from different income classes: 58% (72 out of 124) of low (LICs), and upper/lower middle-income countries (UMCs and LMCs) had less than 0.5% of their cases sequenced in the first 2 years of pandemic, while among HICs, such low levels of surveillance were only observed in 21.5% of the countries (14 out of 65) (Figs. 1 and 2A and Supplementary Data 1 and 5). By comparing the first and second years, however, important increases in sequencing intensity were observed

in HICs, UMCs and LMCs, which expanded their weekly percentage of sequenced cases by 4.7, 15 and 22.5-fold, respectively. For LICs, no major improvements in sequencing intensity were observed (Fig. S5A).

Another key aspect of genomic surveillance is timeliness, which we evaluated by looking at the turnaround time (TAT; defined as the time in days between sample collection and genome submission to GISAID) of SARS-CoV-2 genome sequencing across 19 geographic regions (Fig. 2C; see also ref. 36). We observed that following the detection of more transmissible variants (VOCs) in late 2020, almost all geographic regions decreased their TAT (Fig. 2C and see Fig. S6). Countries in Northern Europe, which had the fastest TAT (Fig. 2C), decreased their median TAT from 20 to 10 days in the second pandemic year. The overall global decrease in TAT also matches a series of bulletins and guidelines for SARS-CoV-2 sequencing, which were published by the WHO and ECDC in early 2021, in the aftermath of the detection of the Alpha VOC[37–40]. In the second pandemic year, we only observed large increases in TATs for Northern and Western Africa (Fig. 2C). When we compare the timeliness of countries based on their income classes, improvements were observed in all classes, except among low income countries, which had higher median TAT in the second pandemic

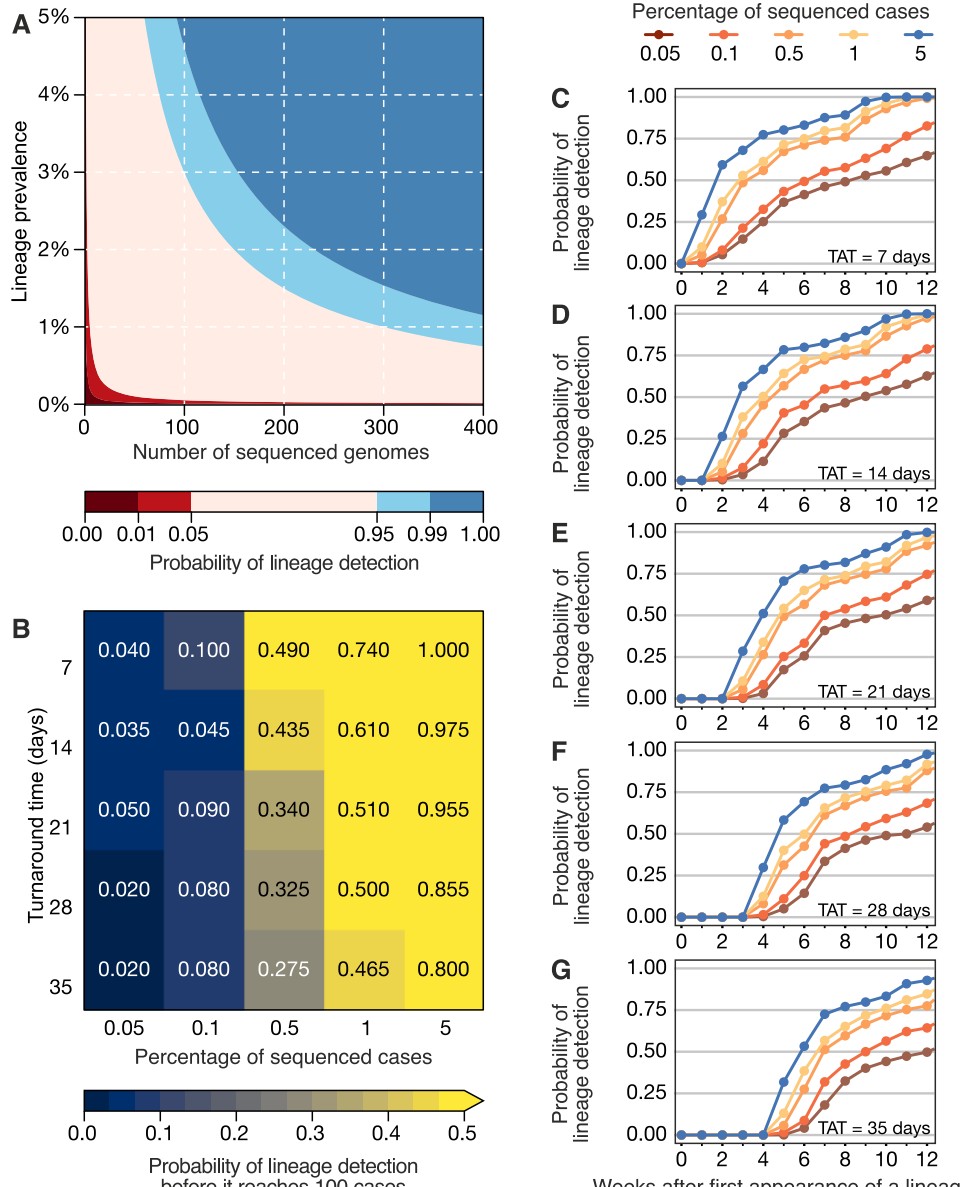

**Fig. 3 | Detection of SARS-CoV-2 lineages under different genomic surveillance scenarios, assuming random sampling. A** The probability of detecting at least one genome of a rare lineage under different sequencing regimes. **B** Relative importance of decreasing genomic sequencing turnaround time (TAT) versus increasing sequencing percentage, measured as the probability that a given lineage

(in simulated datasets) was detected before it had reached 100 cases (described in Fig. S8) across $n = 100$ resamplings. **C–G** Probability of detecting any of the top 10 most prevalent lineages considering TATs of 7, 14, 21, 28 and 35 days across $n = 100$ resamplings.

year (median change from 71 to 109 days of TAT, see Fig. S5B). Rapid generation and sharing of pathogen sequence data from regularly-collected samples is essential to maximize public health impact of genomic data[41,42]. The VOCs Alpha and Gamma, for example, reached up to 50% frequency within 2–3 months of their emergence in the UK and Manaus, respectively[43,44], while with its faster epidemic spread, Omicron took less than a month to reach predominance in South Africa[45]. These examples illustrate that rapid TATs are essential for the early recognition and timely assessments of VOC's transmissibility[41]. The fast detection and characterization of VOCs and VOIs, both in HICs and LMICs, highlights positive examples of how rapid genomic surveillance efforts can aid public health responses, both locally and globally. Genomic surveillance especially in LMICs has provided critical information on the early spread and transmissibility of four novel VOCs (Beta, Gamma, Delta, and Omicron), an important achievement that also set the

foundations for pandemic preparedness in areas that are most at risk for the emergence of zoonotic diseases.

In countries with limited sequencing capacity and/or long TATs, more affordable PCR-based tests, such as RT-PCR tests that distinguish VOCs based on target failures (for example, "S gene target failure"), have been extremely valuable to provide evidence of the spread of a few variants, such as the VOCs Alpha and Omicron, which contain specific deletions that lead to target failures[46]. These tests, however, can only be deployed once enough genomes of a new lineage are sequenced, not only to verify its public health relevance, but also to confirm the presence and high prevalence of unique alleles (with deletions or extensive genetic changes) that allow differential RT-PCR detection. Thus, without rapid sequencing and genomic characterization in the first place, as we observed for Omicron in late 2021[45,46], low-cost PCR-based methods cannot be developed nor deployed.

**Table 1 | Empirical country sequencing capacities at different income levels and lines of inquiry enabled at each level**

| Income class | Median weekly genomes (when sequencing at all) | Mean weekly genomes (when sequencing at all) | Probability of detecting a lineage at 5% prevalence under mean weekly sequencing regime | Maximum probable prevalence of an undetected lineage under mean weekly sequencing regime | Lines of inquiry available |
|---|---|---|---|---|---|
| Low income countries (LICs) | 5 | 10.32 | 0.403 | 0.217 | Presence/absence of prevalent lineages |
| Lower middle-income countries (LMCs) | 10 | 88.92 | 0.988 | 0.028 | + Quantification of lineage prevalence with some error; identification of preliminary patterns of geographic spread |
| Upper middle-income countries (UMCs) | 17 | 147.30 | 0.999 | 0.017 | |
| High-income countries (HICs) | 98 | 1717.01 | 1.000 | 0.001 | + Investigations of lineage dynamics, and transmissibility; high precision lineage tracking (molecular evolution and geographic spread) |

Countries at each income level have markedly different sequencing capacities, allowing for different degrees of epidemic resolution and lines of inquiry. Characteristics of each income class are shown in Supplementary Data 4.

## Sampling strategies for rapid variant detection

We then investigated the impact of genome sequencing intensity and TAT on the detection of SARS-CoV-2 lineages. First, we found that the number of globally observed lineages correlates with the number of SARS-CoV-2 genomes available per country (Pearson's $r = 0.96$, $p$ value < 0.0001) and the overall proportion of sequenced cases in each country (Pearson's $r = 0.51$, $p$ value < 0.0001) (Fig. S7), similar to what has been observed for the UK[47]. This suggests that limited genome sequencing intensity delays the identification and response to new viral lineages with altered epidemiological and antigenic characteristics.

To investigate strategies for rapid variant detection, we simulated the impact of the percentage of sequenced cases and TAT on the reliable detection of previously-identified SARS-CoV-2 lineages using metadata from Denmark, which has one of the most comprehensive SARS-CoV-2 genome surveillance systems (see "Materials and methods", Fig. S8). Here, we assumed a recommended scenario of random sampling, whereby samples for virus genomic sequencing are selected independently of sample metadata such as age, sex, or clinical symptoms[48]. When calculating the probability of detecting at least one genome of a rare lineage (0–5% prevalence) under different sequencing intensities, we found that sequencing at least 300 genomes per week is required to detect, with a 95% probability, a lineage that is circulating in a population at a weekly prevalence of 1%. For a weekly prevalence of 5%, this number decreases to 75 genomes per week (Fig. 3A). These figures are independent of outbreak and population size of a given location, assume representative sampling, and can only tell if a lineage is present, not how prevalent it is. By simulating a scenario of non-random sampling, focused in the most populous region of a country, we observed that the power to detect lineages decreases, but remains moderately useful when TAT is below 21 days, and sequencing intensity is at least 0.5% of all cases (Fig. S9). For other countries, successful detection of domestic lineages from individual regions will also depend on the distribution of population density and human mobility, aspects that are worthy of further investigation in future research. On average, genome surveillance programmes in high income countries should be able to detect circulating virus lineages at 5% prevalence with maximum probability with their current TATs and sequencing intensities, and under the assumption of random sampling (Fig. 3B and Table 1). However, under a scenario of random sampling, low income countries that typically sequence an average of 10 genomes per week may miss a SARS-CoV-2 lineage circulating at up to 21.7% prevalence (Table 1). This will present a substantial limitation to the lines of inquiry available to such countries from genomic sequencing data (Table 1). Within the range of 0.05–5% of sequenced cases considered here, increasing sampling intensity, and to a lesser extent reducing TAT, strongly improves the rapid detection of viral lineages (Fig. 3B).

Next, we simulated 25 scenarios with 100 replicates, in which we varied sampling frequency (from 0.05 to 5%) and TAT (from 7 to 35 days) to compute the probabilities of detecting at least one genome of a given lineage before the lineage reaches a cumulative size of 100 cases (Fig. 3B), using as "ground truth" a dataset from a well characterized setting (see "Materials and methods" and Fig. S8). The simulated scenario shows that when sequencing percentages of 5% per week and TATs of 7 days are achieved in a given setting, a viral lineage is always detected before it reaches 100 cases. When the proportion of sequenced cases per week decreases by 100-fold, to 0.05%, the probability of the timely detection of a viral lineage before it reaches 100 cases decreases to 4% for TATs of 7 days, and further declines to 2.0% when TAT is 35 days (Fig. 3B). These estimates, however, apply to a scenario of random sampling. The power to detect lineages decreases when the sampling is non-random, for example, when focusing only on the most populous region of a country; however, sequencing at least 0.5% of the reported cases with a TAT <21 days remains an important

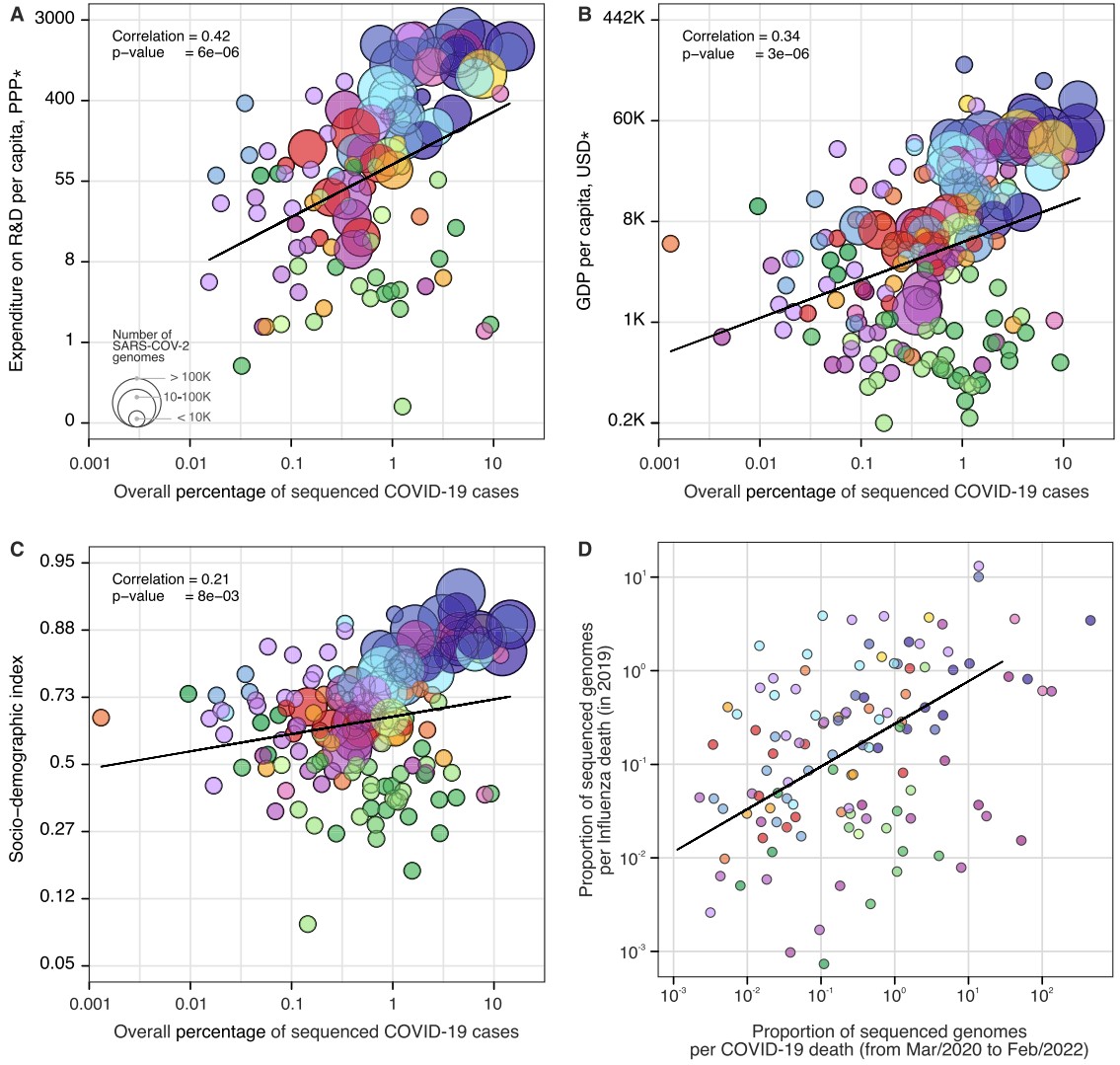

**Fig. 4 | Case sequencing percentages and socioeconomic covariates.** Covariates that show the highest correlation with the overall percentage of COVID-19 sequenced cases (during the period shown in Fig. 1, with geographic regions colored as shown in that figure). **A** Expenditure on R&D per capita (slope = 1.30, CI = (0.76, 1.84), *t*-value = 4.76). **B** GDP per capita (slope = 0.75, CI = (0.44, 1.05), *t*-value = 4.83). **C** Socio-demographic index (slope = 0.25, CI = (0.07, 0.44), *t*-value = 2.70). **D** Overall proportion of sequenced genomes per influenza death in 2019 (HA segment) (slope = 0.89, CI = (0.40, 1.37), *t*-value = 3.62). For correlations between covariates and turnaround time, see Fig. S10. The color scheme is the same as in Figs. 1 and 2. Solid line shows the linear fit; correlation is Pearson's correlation; *p* values are reported based on the *t*-statistic using two-sided hypothesis, with the null hypothesis being that the slope of the linear trend is zero. There was no need for multiple comparison adjustments. *PPP purchasing power parity, USD US dollar 2005.

factor in successful detection even in non-random sampling scenarios (Fig. S9).

For an optimistic scenario of 0.5% sequenced cases (achieved by 78% HICs and 40% LMICs) and a TAT of 21 days (observed in 25% of the genomes submitted by HICs, and in 5% by LMICs) (Supplementary Data 4), we found a 34% probability of detecting a lineage before it reaches 100 cases. Throughout the pandemic, many countries reported weekly incidences as high as 100 cases per 100,000 inhabitants (Figs. 1C and S3 and S4). For example, in a scenario of high incidence, for Manaus, a city with 2.2 million inhabitants in the Amazonas state located in the North of Brazil, the 0.5% sequencing threshold would correspond to 11 randomly selected genomes per week. With a 21-day TAT, this would allow the detection of a given lineage with a 34% probability (Fig. 3B). For São Paulo city (12.4 million inhabitants), this number increases to 62 genomes per week. For Brazil (212.6 million inhabitants), this would correspond to 1063 weekly genomes selected from a random population of samples, in the above mentioned scenario of high incidence. Although the 0.5% ratio of sequenced cases

per week in near real-time is a reasonable benchmark for SARS-CoV-2 genomic surveillance in 78% of high income countries (Supplementary Data 4), this often comes as a result of close coordination between diagnostic centers and well-funded, decentralized infrastructures to integrate sequencing data and sample-associated metadata (see e.g. ref. 49).

### Factors associated with genomic surveillance capacity

While many HICs were able to rely on previously established networks and laboratory infrastructure to perform molecular testing and sequencing[50,51], many LMICs—including Brazil, South Africa, and India where four VOCs were first detected[43,52–54]—have faced additional challenges to the rapid expansion of genomic surveillance[51,55,56]. Pathogen genomics complements but often competes for limited resources with other aspects of pandemic response, for instance, surveillance and testing capacity, medical supplies, laboratory reagents, public health and social measures and vaccine development[57]. To investigate how socioeconomic factors can impact

SARS-CoV-2 genomic surveillance response around the world, we explored the correlation between the percentage of sequenced COVID-19 cases in each country, and 20 country-level socioeconomic and health quality covariates (Fig. 4 and Supplementary Data 5). We found that the percentage of sequenced cases is significantly associated with expenditure on research and development (R&D) per capita ($r = 0.47$, $p$ value <0.0001) (Fig. 4A), gross domestic product (GDP) per capita ($r = 0.37$, $p$ value <0.0001) (Fig. 4B), socio-demographic index ($r = 0.31$, $p$ value <0.001) (Fig. 4C), and established influenza virus genomic surveillance capacity prior to the COVID-19 pandemic ($r = 0.30$, $p$ value <0.001) (Fig. 4D and Supplementary Data 6).

A total of 74% (140 out of 189) of the countries that submitted SARS-CoV-2 genomes to GISAID had also shared influenza virus sequences to that same database in 2019. When compared by income class, we observed that the majority of UMCs (77%) and HICs (78%) currently sequencing SARS-CoV-2 had already reported influenza virus sequences in public databases up to 2019. For LIC countries, this drops to 37.5%, suggesting that many LICs initiated or enhanced their genome sequencing programs during the COVID-19 pandemic. While disparities in investment in national health, research, and development continue to impact the ability of countries to scale up genomic sequencing intensity[28,51,58], recent improvements in genomic surveillance by many LMICs (Fig. S5) and the association of sequencing efforts with established genomic surveillance capacity paint an encouraging picture for future pandemic preparedness programs.

When we explored correlations with mean TAT (Supplementary Data 7), we found that healthcare access and quality index ($r = -0.56$, $p$ value <0.0001), universal health coverage ($r = -0.56$, $p$ value <0.0001), health worker density ($r = -0.56$, $p$ value <0.0001), and health expenditure per capita ($r = -0.54$, $p$ value <0.0001) are significantly correlated with mean TATs (Fig. S10 and Supplementary Data 7). Our results quantify only correlations between socioeconomic covariates, sequencing intensity, and TAT, and cannot be interpreted as causal. Future studies should focus on additional variables that may affect genomic surveillance, especially in LMICs, such as training laboratory and bioinformatic personnel, metadata standards, costs associated with imported consumables, and shipment delays that may be exacerbated by border closures and travel restrictions[28,55,56,58,59]. Other factors associated with delays in reporting VOCs include social and political stigma and perceived negative impact on travel when reporting potential VOCs, and concerns of having findings scooped and published by other researchers[60]. Longer TATs are also expected in countries where virus genomics activities are focused on retrospective genomic studies to investigate SARS-CoV-2 reinfections[61], vaccine breakthrough infections[62], and past epidemic dynamics[63,64].

## Discussion

Leveling up pathogen genomic surveillance efforts, particularly in LMICs, should be a priority to improve pandemic preparedness worldwide[60]. Our findings demonstrate that global SARS-CoV-2 genomic surveillance efforts are currently highly unbalanced, and contingent upon socioeconomic factors and pre-pandemic laboratory and surveillance capacity. Our results suggest that sequencing 0.5% of total confirmed cases, with a TAT below 21 days, could provide a benchmark for genomic surveillance studies targeting SARS-CoV-2 and future emerging viruses. Alongside with the guidance provided by the WHO and other international public health authorities (see[37,38,40,65–69]), ongoing surveys to understand barriers to virus genome sequencing and sampling selection strategies will provide valuable information for future surveillance programs. Implementation of metagenomic approaches for virus discovery followed by virus-genome specific sequencing approaches could help overcome existing limitations of molecular and syndromic surveillance strategies[70]. Adoption of standardized protocols for

representative genomic surveillance strategies[40,48], establishment of data and minimal metadata standards, efficient and facilitated access to information, following equitable data sharing agreements[65], and collaboration between academia, public health laboratories, private laboratories and other stakeholders will be essential to maximize cost-effectiveness and public health impact of genomic surveillance. While a random sampling strategy may provide accurate information into SARS-CoV-2 variant emergence and frequency estimation, we note that genome sampling strategies should be considered pathogen- and question-specific[48,65,66]. For example, non-random selection of samples stratified by disease severity may be required to identify genes or mutations associated with clinical outcomes[71].

There are several global efforts underway to improve genomic sequencing capacities around the world, including the AFRO-Africa Centre for Disease Control, the Pan American Health Organization COVIGEN Network, Regional Genomic Surveillance Consortium from WHO Southeast Asia Region, and the ACT-A WHO Global Risk Monitoring Framework. Global efforts must be made to improve in-country genomic surveillance capacity, and to provide sustainable research funding for strengthening sequencing capacity and outbreak analytics, particularly in LMICs. Improved pathogen surveillance at the human, animal and human-animal interfaces is also urgently needed[72]. Retaining existing and expanding local capacity efforts acquired during the SARS-CoV-2 pandemic will be critical to contain and respond to the next "Disease X"[72].

## Methods

### Genomic surveillance and epidemiological data

To obtain the percentage of sequenced cases for each country, per week and cumulative, we used metadata related to the "country of exposure" of genomes submitted to GISAID[30] up to March 18th, 2022, collected from EW (epidemiological week) 10 of 2020 (March 1st, 2020) to EW 8 of 2022 (February 26th, 2022). We obtained global daily COVID-19 case counts from Johns Hopkins University, Center for Systems Science and Engineering (http://github.com/CSSEGISandData/COVID-19), and population data from each country from the United Nations' Department of Economic and Social Affairs[73]. Countries were grouped by income using the current classification by the World Bank[74]. We calculated weekly percentages of COVID-19 cases sequenced per country by aggregating and dividing genome and case counts per EW, using a custom pipeline "subsampler" (http://github.com/andersonbrito/subsampler)[75].

### Analysis of covariates correlated with genomic surveillance capacity

Covariates related to health systems were available from the Institute for Health Metrics and Evaluation (IHME)[76], GDP data were also available from IHME[77], and data on R&D expenditure per capita were available from UNESCO[78]. For the covariates from IHME[76] we have selected their values for the year 2019, for GDP data for the year 2015, and for R&D expenditure we calculated country-level means for the years 2013 through 2019. Influenza virus genomic data (HA segment) collected in 2019 were obtained from GISAID[30], and 2019 influenza death estimate data were downloaded from the IHME Global Burden of Disease Study 2019[76]. Correlations and covariate details are provided in Supplementary Data 5. To calculate correlations, the percentage of sequenced cases was $\log_{10}$-transformed. Transformations applied to covariates are provided in Supplementary Data 5, in column "transformation". For each covariate we have estimated a linear fit by applying a generalized linear model, regressing a covariate (possibly, transformed, as indicated in Supplementary Data 6) onto the $\log_{10}$-transformed percentage of sequenced cases; $p$ values corresponding to the estimated slopes are available in Figs. 3 and S10.

## Simulation of scenarios of genome sampling

As shown in Fig. 1, Denmark has one of the most comprehensive genomic surveillance programs in this COVID-19 pandemic, sequencing around 14.5% of its reported cases up to February 26th, 2022 (2,733,807 cases and 396,994 genomes with >70% coverage; access date: March 18th, 2022)[79]. In order to simulate the impact of the percentage of sequenced cases and the TAT (time between sample collection and genome submission) in the detection of previously-identified SARS-CoV-2 lineages in a given country, we used metadata from genomes obtained by the Danish COVID-19 genome consortium, with collection dates between EW 10 of 2020 (March 1st) and EW 8 of 2022 (February 26th)[79].

To evaluate the impact of temporal delays between reported dates of sample collection and dates of genome submission on GISAID, we generated lists of genomes with adjusted submission dates, to simulate TAT representing delays between 7 and 35 days (5 weeks) between sample collection and genome submission. Considering the high percentage of sequenced cases per EW in Denmark (often above 20%), we produced several genome datasets by simulating scenarios with different percentages of sequenced cases per EW (0.05, 0.1, 0.5, 1 and 5%). In doing so we were able to simulate 25 scenarios (with 100 replicates each) with combinations of different TAT and percentage of sequenced cases in order to assess how these two parameters may impact our ability (expressed as a probability) to detect circulating lineages. Specifically, we randomly sampled each column of the observed data (considered them to be case counts across all circulating lineages) according to the targeted percentage of sequenced cases which would become available after a given TAT, ignoring rare lineages that never reached 100 sampled genomes. Each combination of percentage of sequenced cases and TAT yielded one table of genomes available across the EWs. This procedure was repeated 100 times to mitigate random sampling effects, and results were used to generate a probability of detection for each circulating lineage. Summarizing the 100 replicates led to detection probabilities for each lineage in each epidemiological week. To simulate uneven geographic distribution of sequenced cases, we also simulated an analogous scenario to the one described above but where only the sequencing intensity in Hovedstaden, Denmark's capital region, was used in simulations and compared to actual lineage frequency data for all of Denmark (Fig. S9). Figure 3A shows the probability of not drawing 0 from a Poisson distribution whose mean is the product of lineage prevalence and sequenced cases. In Fig. 3B, we show the computed probabilities of detection across simulation replicates, at a given sampling frequency and delay, which were able to have at least one detection of a given lineage before reaching a cumulative size of 100 cases in the full dataset without delays ("ground truth", see Fig. S8). Figure 3C–G similarly map this out, but in time, asking how long it takes for a given lineage to be detected over time using the first instance of a lineage in the "ground truth" dataset as its emergence.

### Reporting summary

Further information on research design is available in the Nature Research Reporting Summary linked to this article.

## Data availability

The findings of this study are based on metadata associated with 8,949,097 sequences available on GISAID up to March 18th, 2022, and accessible at https://doi.org/10.55876/gis8.220330me. Epidemiological data of global reported cases were downloaded from the GitHub account of the CSSE at Johns Hopkins University (https://github.com/CSSEGISandData/COVID-19). All relevant data used in this study are available as Supplementary files in this manuscript, and on the following GitHub repository: https://github.com/andersonbrito/paper_2022_metasurveillance.

## Code availability

The pipeline used to calculate the percentages of sequenced cases per country is available on the following GitHub repository: https://github.com/andersonbrito/subsampler[75].

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

## Acknowledgements

We gratefully acknowledge the authors from the originating laboratories responsible for obtaining the specimens, as well as the submitting laboratories where the genomic data were generated and shared via GISAID, on which this research is based. An acknowledgment table can be found in Supplementary Data 8 and at gisaid.org with set accession EPI_SET_20220330me. We thank James Nokes, Isabella Lynette Ochola, and Sylvie Briand for their valuable comments. G.D. acknowledges Joshua Batson, whose work shared on Twitter (@thebasepoint) inspired the creation of Fig. 3A. E.S. and S.F. acknowledge the EPSRC (EP/V002910/1). G.B. acknowledges support from the Internal Funds KU Leuven (Grant No. C14/18/094) and the Research Foundation—Flanders ("Fonds voor Wetenschappelijk Onderzoek—Vlaanderen," G0E1420N, G098321N). G.W.H. acknowledges support from NIH F31 AI154824. M.A.S. acknowledges support from grants NIH R01 AI153044 and NIH U19 AI135995. M.U.G.K. acknowledges funding from the Oxford Martin School, EUH2020 project MOOD, Branco Weiss Fellowship and grants from The Rockefeller Foundation and Google.org. N.D.G. acknowledges support from Fast Grant from Emergent Ventures at the Mercatus Center at George Mason University and CDC Contract # 75D30120C09570. O.G.P. acknowledges support from the Oxford Martin School. N.R.F. acknowledges support by a Wellcome Trust and Royal Society Sir Henry Dale Fellowship (204311/Z/16/Z); acknowledges funding from the MRC Centre for Global Infectious Disease Analysis (reference MR/R015600/1), jointly funded by the UK Medical Research Council (MRC) and the UK Foreign, Commonwealth & Development Office (FCDO), under the MRC/FCDO Concordat agreement and is also part of the EDCTP2 programme supported by the European Union; and acknowledges funding by Community Jameel. N.R.F. and E.C.S. acknowledge support by the Medical Research Council-São Paulo Research Foundation (FAPESP) CADDE partnership award (MR/S0195/1 and FAPESP 18/14389-0) (http://caddecentre.org/) and Bill & Melinda Gates Foundation (INV-034540 and INV-034652). Rede Corona-ômica BR MCTI/FINEP is affiliated to RedeVírus/MCTI (awards FINEP = 01.20.0029.000462/20, CNPq = 404096/2020-4). B.P.H. and V.S. acknowledge the contribution of SARS-CoV-2 genomes by members of the Communicable Diseases Genomics Network of Australia. C.C.K. acknowledges support from the US Public Health Service Ruth L. Kirschstein National Research Service Award (5T35HL007649-35). R.S.A. acknowledges funding from CNPq: 312688/2017-2 and 439119/2018-9; MEC/CAPES: 14/2020—23072.211119/2020-10; FINEP: 0494/20 01.20.0026.00 and UFMG-NB3 1139/20 and FAPERJ: 202.922/2018. A.T.R.V. acknowledges the Corona-ômica-RJ (FAPERJ E-26/210.179/2020 and E-26/211.107/2021) and CNPq (307145/2021-2 and 440931/2020-7). I.A., I.N.I. and I.P. acknowledge a grant from the Ministry of Education and Science, Bulgaria (contract: КП-06-Н43/1-27.11.2020). S.C.H. acknowledged support by a Wellcome Trust Sir Henry Wellcome Fellowship (220414/Z/20/Z).

## Author contributions

Conception: A.F.B., N.D.G., G.B., and N.R.F.; Data acquisition: Bulgarian SARS-CoV-2 sequencing group, Communicable Diseases Genomics Network (Australia and New Zealand), COVID-19 Impact Project, Danish Covid-19 Genome Consortium, Fiocruz COVID-19 Genomic Surveillance Network, GISAID core curation team, Network for Genomic Surveillance in South Africa (NGS-SA), Swiss SARS-CoV-2 Sequencing Consortium, G.G., C.N.A., R.T.P.L., M.M.S., P.C.R., C.V.F.C., N.S.D.S., S.M.S.; Analysis: A.F.B., E.S., G.D., G.W.H., C.C.K., J.H., S.M.S., S.B., S.F., M.A.S., and G.B.; Interpretation: A.F.B., E.S., G.D., G.W.H., M.U.G.K., J.H., H.T., G.G., C.N.A., T.d.O., R.T.P.L., S.M.S., S.C.H., O.G.P., C.D., S.B., S.F., N.D.G., G.B., and N.R.F.; Drafting: A.F.B., E.S., G.D., C.C.K., G.B., and N.R.F.; Revising: A.F.B., E.S., G.D., G.W.H., M.U.G.K., J.H., H.T., G.G., C.N.A., L.E.M., C.W., B.P.H., V.S., N.S.Z., O.M., H.M.B., T.d.O., R.T.P.L., M.M.S., P.C.R., A.T.R.V., F.R.S., R.S.A., I.A., I.N.I., I.P., C.V.F.C., N.S.D.S., B.B., C.G., S.M.S., D.N., K.v.E. M.P., M.v.K., S.C.H., E.C.S., O.G.P., C.D., M.A.S., N.D.G., G.B., and N.R.F.; Funding: N.D.G. and N.R.F.

## Competing interests

N.D.G. is an infectious diseases consultant for Tempus Labs and the National Basketball Association. M.A.S. receives grants and contracts from the National Institutes of Health, the US Food & Drug Administration, the US Department of Veterans Affairs and Janssen Research & Development. O.G.P. has undertaken work for AstraZeneca on SARS-CoV-2 classification and genetic lineage nomenclature. The remaining authors declare no competing interests.

## Additional information

**Correspondence and requests** for materials should be addressed to Anderson F. Brito or Nuno R. Faria.

Anderson F. Brito [1,2,71] ✉, Elizaveta Semenova [3,71], Gytis Dudas [4,71], Gabriel W. Hassler[5], Chaney C. Kalinich[1,6], Moritz U. G. Kraemer [7], Joses Ho [8,9], Houriiyah Tegally[10,11], George Githinji [12,13], Charles N. Agoti [12,14], Lucy E. Matkin [7], Charles Whittaker[15,16], Bulgarian SARS-CoV-2 sequencing group*, Communicable Diseases Genomics Network (Australia and New Zealand)*, COVID-19 Impact Project*, Danish Covid-19 Genome Consortium*, Fiocruz COVID-19 Genomic Surveillance Network*, GISAID core curation team*, Network for Genomic Surveillance in South Africa (NGS-SA)*, Swiss SARS-CoV-2 Sequencing Consortium*, Benjamin P. Howden [17], Vitali Sintchenko [18,19], Neta S. Zuckerman[20], Orna Mor [20], Heather M. Blankenship[21], Tulio de Oliveira [10,11,22,23], Raymond T. P. Lin [24], Marilda Mendonça Siqueira[25], Paola Cristina Resende [25], Ana Tereza R. Vasconcelos [26], Fernando R. Spilki[27], Renato Santana Aguiar [28,29], Ivailo Alexiev[30], Ivan N. Ivanov[30], Ivva Philipova[30], Christine V. F. Carrington[31], Nikita S. D. Sahadeo[31], Ben Branda[8], Céline Gurry[8], Sebastian Maurer-Stroh[8,9,24], Dhamari Naidoo[32], Karin J. von Eije [33,34], Mark D. Perkins[34], Maria van Kerkhove [34], Sarah C. Hill[35], Ester C. Sabino[2,36], Oliver G. Pybus [7,35], Christopher Dye[7], Samir Bhatt[15,16,37], Seth Flaxman [3], Marc A. Suchard [5,38,39], Nathan D. Grubaugh[1,40,72], Guy Baele [41,72] & Nuno R. Faria [7,15,16,36,72] ✉

[1]Department of Epidemiology of Microbial Diseases, Yale School of Public Health, New Haven, CT, USA. [2]Instituto Todos pela Saúde, São Paulo, SP, Brazil. [3]Department of Computer Science, University of Oxford, Oxford, UK. [4]Institute of Biotechnology, Life Sciences Center, Vilnius University, Vilnius, Lithuania. [5]Department of Computational Medicine, David Geffen School of Medicine, University of California Los Angeles, Los Angeles, CA, USA. [6]Yale School of Medicine, Yale University, New Haven, CT, USA. [7]Department of Biology, University of Oxford, Oxford, UK. [8]GISAID Global Data Science Initiative, Munich, Germany. [9]Bioinformatics Institute & ID Labs, Agency for Science Technology and Research, Singapore, Singapore. [10]KwaZulu-Natal Research Innovation and Sequencing Platform (KRISP), School of Laboratory Medicine and Medical Sciences, University of KwaZulu-Natal, Durban, South Africa. [11]Centre for Epidemic Response and Innovation (CERI), School of Data Science and Computational Thinking, Stellenbosch University, Stellenbosch, South Africa. [12]KEMRI-Wellcome Trust Research Programme, Kilifi, Kenya. [13]Department of Biochemistry and Biotechnology, Pwani University, Kilifi, Kenya. [14]School of Health and Human Sciences, Pwani University, Kilifi, Kenya. [15]MRC Centre for Global Infectious Disease Analysis, School of Public Health, Imperial College London, London, UK. [16]The Abdul Latif Jameel Institute for Disease and Emergency Analytics (J-IDEA), School of Public Health, Imperial College London, London, UK. [17]Microbiological Diagnostic Unit Public Health Laboratory, Department of Microbiology and Immunology, The University of Melbourne at The Peter Doherty Institute for Infection and Immunity, Melbourne, VIC, Australia. [18]Sydney Institute for Infectious Diseases, The University of Sydney, Sydney, NSW, Australia. [19]Institute of Clinical Pathology and Medical Research, NSW Health Pathology, Westmead, NSW, Australia. [20]Central Virology Laboratory, Israel Ministry of Health, Sheba Medical Center, Ramat Gan, Israel. [21]Michigan Department of Health and Human Services, Bureau of Laboratories, Lansing, MI, USA. [22]Centre for the AIDS Programme of Research in South Africa (CAPRISA), Durban, South Africa. [23]Department of Global Health, University of Washington, Seattle, WA, USA. [24]National Centre for Infectious Diseases, Singapore, Singapore. [25]Laboratory of Respiratory Viruses and Measles, Instituto Oswaldo Cruz, FIOCRUZ, Rio de Janeiro, Brazil. [26]Laboratório de Bioinformática, Laboratório Nacional de Computação Científica, Petrópolis, Brazil. [27]Feevale University, Institute of Health Sciences, Novo Hamburgo, RS, Brazil. [28]Laboratório de Biologia Integrativa, Departamento de Genética, Ecologia e Evolução, Instituto de Ciências Biológicas, Universidade Federal de Minas Gerais, Belo Horizonte, Brazil. [29]Instituto D'Or de Pesquisa e Ensino (IDOR), Rio de Janeiro, Brazil. [30]National Center of Infectious and Parasitic Diseases, Sofia, Bulgaria. [31]Department of Preclinical Sciences, Faculty of Medical Sciences, The University of the West Indies, St. Augustine, Trinidad and Tobago. [32]Health Emergencies Programme, World Health Organization Regional Office for South-East Asia, New Delhi, India. [33]Department of Medical Microbiology and Infection Prevention, Division of Clinical Virology, University of Groningen, University Medical Center Groningen, Groningen, The Netherlands. [34]Emerging Diseases and Zoonoses Unit, Health Emergencies Programme, World Health Organization, Geneva, Switzerland. [35]Royal Veterinary College, Hawkshead, UK. [36]Instituto de Medicina Tropical, Faculdade de Medicina da Universidade de São Paulo, São Paulo, Brazil. [37]Section of Epidemiology, Department of Public Health, University of Copenhagen, Copenhagen, Denmark. [38]Department of Biostatistics, Fielding School of Public Health, University of California Los Angeles, Los Angeles, CA, USA. [39]Department of Human Genetics, David Geffen School of Medicine, University of California Los Angeles, Los Angeles, CA, USA. [40]Department of Ecology and Evolutionary Biology, Yale University, New Haven, CT, USA. [41]Department of Microbiology, Immunology and Transplantation, Rega Institute, KU Leuven, Leuven, Belgium. [71]These authors contributed

equally: Anderson F. Brito, Elizaveta Semenova, Gytis Dudas. [72]These authors jointly supervised this work: Nathan D. Grubaugh, Guy Baele, Nuno R. Faria. *Lists of authors and their affiliations appear at the end of the paper. ✉e-mail: andersonfbrito@gmail.com; nfaria@ic.ac.uk

## Bulgarian SARS-CoV-2 sequencing group

Todor Kantardjiev[30], Nelly Korsun[30], Savina Stoitsova[30], Reneta Dimitrova[30], Ivelina Trifonova[30], Veselin Dobrinov[30], Lubomira Grigorova[30], Ivan Stoykov[30], Iliana Grigorova[30], Anna Gancheva[30], Ivan N. Ivanov[30], Ivva Philipova[30] & Ivailo Alexiev[30]

## Communicable Diseases Genomics Network (Australia and New Zealand)

Amy Jennison[17], Lex Leong[17], David Speers[17], Rob Baird[17], Louise Cooley[17], Karina Kennedy[17], Joep de Ligt[17], William Rawlinson[17], Sebastiaan van Hal[17], Deborah Williamson[17], Vitali Sintchenko [18,19] & Benjamin P. Howden [17]

## COVID-19 Impact Project

Risha Singh[42], SueMin Nathaniel-Girdharrie[42], Lisa Edghill[42], Lisa Indar[42], Joy St. John[42], Gabriel Gonzalez-Escobar[42], Vernie Ramkisoon[31], Arianne Brown-Jordan[31], Anushka Ramjag[31], Nicholas Mohammed[31], Jerome E. Foster[31], Irad Potter[43], Sharra Greenaway-Duberry[44], Kenneth George[45], Sharon Belmar-George[46], John Lee[47], Jacqueline Bisasor-McKenzie[48], Nadia Astwood[49], Rhonda Sealey-Thomas[50], Hazel Laws[51], Narine Singh[52], Ayoola Oyinloye[53], Pearl McMillan[54], Avery Hinds[55], Naresh Nandram[55], Roshan Parasram[55], Zobida Khan-Mohammed[55], Shawn Charles[56], Aisha Andrewin[57], David Johnson[58], Simone Keizer-Beache[59], Chris Oura[60], Sarah C. Hill[35], Oliver G. Pybus[35,61], Nuno R. Faria[15,61], Nikita S. D. Sahadeo[31] & Christine V. F. Carrington[31]

[42]Caribbean Public Health Agency, Port of Spain, Republic of Trinidad and Tobago. [43]Ministry of Health and Social Development, Road Town, Tortola, British Virgin Islands. [44]Ministry of Health and Social Services, Brades, Montserrat. [45]Ministry of Health and Wellness, Bridgetown, Barbados. [46]Ministry of Health and Wellness, Castries, Saint Lucia. [47]Ministry of Health and Wellness, George Town, Cayman Islands. [48]Ministry of Health and Wellness, Kingston, Jamaica. [49]Ministry of Health, Agriculture, Sports and Human Services, Cockburn Town, Turks and Caicos Islands. [50]Ministry of Health, St John's, Antigua and Barbuda. [51]Ministry of Health, Basseterre, Saint Kitts and Nevis. [52]Ministry of Health, Georgetown, Guyana. [53]Ministry of Health, Hamilton, Bermuda. [54]Ministry of Health, Nassau, Bahamas. [55]Ministry of Health, Port of Spain, Republic of Trinidad and Tobago. [56]Ministry of Health, St. Georges, Grenada. [57]Ministry of Health, The Valley, Anguilla. [58]Ministry of Health, Wellness and New Health Investment, Roseau, Dominica. [59]Ministry of Health, Wellness and the Environment, Kingstown, Saint Vincent and the Grenadines. [60]School of Veterinary Medicine, Faculty of Medical Sciences, The University of the West Indies, St. Augustine, Republic of Trinidad and Tobago. [61]Department of Zoology, University of Oxford, Oxford, UK.

## Danish Covid-19 Genome Consortium

Marc Stegger[62], Mads Albertsen[63], Anders Fomsgaard[63] & Morten Rasmussen[63]

[62]Aalborg University, Aalborg, Denmark. [63]Statens Serum Institut, Copenhagen, Denmark.

## Fiocruz COVID-19 Genomic Surveillance Network

Ricardo Khouri[64], Felipe Naveca[65], Tiago Graf[64], Fábio Miyajima[66], Gabriel Wallau[67], Fernando Motta[25], Paola Cristina Resende [25] & Marilda Mendonça Siqueira[25]

[64]Oswaldo Cruz Foundation, Salvador, BA, Brazil. [65]Oswaldo Cruz Foundation, Manaus, AM, Brazil. [66]Oswaldo Cruz Foundation, Eusébio, CE, Brazil. [67]Oswaldo Cruz Foundation, Recife, PE, Brazil.

## GISAID core curation team

Shruti Khare[8], Lucas Freitas[8], Constanza Schiavina[8], Gunter Bach[8], Mark B. Schultz[8], Yi Hong Chew[8], Meera Makheja[8], Priscila Born[8], Gabriela Calegario[8], Sofia Romano[8], Juan Finello[8], Amadou Diallo[8], Raphael T. C. Lee[8], Ya Ni Xu[8], Winston Yeo[8], Suma Tiruvayipati[8], Shilpa Yadahalli[8], Joses Ho [8,9], Ben Branda[8], Céline Gurry[8] & Sebastian Maurer-Stroh[8,9,24]

## Network for Genomic Surveillance in South Africa (NGS-SA)

Eduan Wilkinson[10], Arash Iranzadeh[10], Jennifer Giandhari[10], Deelan Doolabh[10], Sureshnee Pillay[10], Upasana Ramphal[10], James E. San[10], Nokukhanya Msomi[10], Koleka Mlisana[10], Anne von Gottberg[10], Sibongile Walaza[10], Arshad Ismail[10], Thabo Mohale[10], Susan Engelbrecht[10], Gert Van Zyl[10], Wolfgang Preiser[10], Alex Sigal[10], Diana Hardie[10], Gert Marais[10], Marvin Hsiao[10], Stephen Korsman[10], Mary-Ann Davies[10], Lynn Tyers[10], Innocent Mudau[10], Denis York[10], Caroline Maslo[10], Dominique Goedhals[10], Shareef Abrahams[10], Oluwakemi Laguda-Akingba[10], Arghavan Alisoltani-Dehkordi[10], Adam Godzik[10], Constantinos K. Wibmer[10], Darren Martin[10], Richard J. Lessells[10], Jinal N. Bhiman[10], Carolyn Williamson[10], Houriiyah Tegally[10,11] & Tulio de Oliveira[10,11,22,23]

## Swiss SARS-CoV-2 Sequencing Consortium

Chaoran Chen[68], Sarah Nadeau[68], Louis du Plessis[68], Christiane Beckmann[69], Maurice Redondo[69], Olivier Kobel[69], Christoph Noppen[69], Sophie Seidel[69], Noemie Santamaria de Souza[69], Niko Beerenwinkel[68], Ivan Topolsky[68], Philipp Jablonski[68], Lara Fuhrmann[68], David Dreifuss[68], Katharina Jahn[68], Pedro Ferreira[68], Susana Posada-Céspedes[68], Christian Beisel[68], Rebecca Denes[68], Mirjam Feldkamp[68], Ina Nissen[68], Natascha Santacroce[68], Elodie Burcklen[68], Catharine Aquino[68], Andreia Cabral de Gouvea[68], Maria Domenica Moccia[68], Simon Grüter[68], Timothy Sykes[68], Lennart Opitz[68], Griffin White[68], Laura Neff[68], Doris Popovic[68], Andrea Patrignani[68], Jay Tracy[68], Ralph Schlapbach[68], Emmanouil Dermitzakis[70], Keith Harshman[70], Ioannis Xenarios[70], Henri Pegeot[70], Lorenzo Cerutti[70], Deborah Penet[70] & Tanja Stadler[68]

[68]ETH Zürich, Zurich, Switzerland. [69]Violler AG, Allschwil, Switzerland. [70]Health 2030 Genome Center, Geneva, Switzerland.

