## [Peer Review File · Nature Communications]

Global disparities in SARS-CoV-2 genomic surveillanceREVIEWER COMMENTS

Reviewer #1 (Remarks to the Author):

Brito et al present a formal analysis of the disparities in SARS-CoV-2 sequencing capacity. This is an interesting and important analysis that presents the current disparity in global SARS-CoV-2 sequencing.

I think the analysis are sound and the paper will be of great use to the scientific, and public health communities and will provide policy makers with useful data.

A major comment is that the authors analyse the problem and provide clear evidence of the inequalities in sequencing and suggest that "We found that sequencing at least 0.5% of the cases, with a TAT <21 days, could be a benchmark for SARS-CoV-2 genomic surveillance efforts." But that is the limit of the recommendations for the future. I would suggest that this paper is an opportunity to present a vision of what a basic sequencing capacity every country should be aiming to have, with details about TAT, geographical coverage, data release, sample types, etc. A figure / flow chart summarising this would be useful.

I have outlined some comments below that I think would improve the paper.

1. Line 134 to 135 – there is no mention of B.1.1.529/Omicron?
2. Line 142 – the last date seems a bit out of date? (I presume this reflects an initial submission a while ago). Could the authors update the numbers to be more contemporary?
3. Figure 1 – (A) There is no key to the colours in Figure 1? Other than the countries highlighted?
4. Figure 1 panel. Could Figure 1 be split into two separate figures? I found it hard to extract information from the figures as the panels were so small. It is a pity because they are such great figures - it would be better if the reader can view them properly. Particularly figure 1A – readers need to see the details. Perhaps this could be Figure 1 and the rest of the panel (B, C, D) would move into a separate figure 2?
5. Line 188 to 191: I found this sentence particularly striking (and shocking). I would suggest that the authors include this point in the Abstract - as it would seem to be a clear objective that could, and should be solved in the immediate future.
6. Line 225 – 227: The authors state here that these figures are independent of outbreak and population size. Did the authors attempt to include considerations of the physical size of the country, population densities and degree of interconnectivity between locations? For example – if a country was to heavily bias sampling in a capital city but then not sample in a secondary cities or locations (that might be many of thousands of kilometres away) and/or be more connected (in terms of transport) to a second country? It may well be that this kind of modelling is beyond the scope of this analysis, but at the least the authors should consider including some discussion around ensuring random sampling attempts to provide broad geographical coverage throughout a country rather than just focusing on one or two locations may risk an inaccurate picture of the true epidemiology. Of course, with the caveat that in resource limited situations something is better than nothing.
7. Line 243 to 251: An additional analysis that wasn't conducted by the authors but would have added to this analysis - is sample selection. For example, in many locations sequencing capacity is often linked to healthcare in which patients often take a number of days or even weeks before being hospitalised with COVID-19 (and other infections). The usefulness and timeliness of sequencing to detect an emerging variant would likely be affected if samples were always taken from hospitalised patients - rather than catching the leading edge of the epidemic by sampling the most recent community samples. Some discussion along these lines should be included.

7. Line 326 - 338:

(a) The authors didn't include any issue around metadata and data deposition in public databases (e.g. the risk of release of patient identifiable data – particularly from populations who are more sensitive to data release and risks of deductive disclosure in remote communities) and then subsequent access to data. This has caused a major delay in the public release of data in a number of countries. Perhaps some comment along the lines that it is more important that data is deposited with a minimal data set (sampling date and country) rather than chasing perfection (e.g. more granular patient and geographical data).

(b) Also, restrictions on data access to data in GISAID has also caused problems. While I don't expect authors to specifically comment on GISAID directly (I note GISAID is included as an author) - a broader discussion of the importance of data sharing (timely and open access with respect to authorship of data generators) would be worthwhile – during the COVID-19 pandemic things are much better – there is still room for improvement.

8. Line 345 to 347 – following on from point 6 – some discussion here about ensuring broad geographical coverage should be included.

9. Line 357 – 358: A bit more detail should be added here about other non-random sample selections including: (a) treatment failures (antivirals), (b) vaccine breakthrough, (c) reinfections, (d) clinical trials, (e) monitoring of immunocompromised infections, etc.

Reviewer #2 (Remarks to the Author):

The paper of Brito and colleagues surveys and analyses the intensity of genomic surveillance for SARS-CoV-2 across the globe. They correlate between sequencing incidence and various features of a country (socioeconomic factors and COVID-19 incidence). They further simulate how many sequences are needed to detect a given variant present at different frequencies.

All in all, this is a very interesting and highly relevant topic that has never been explored before at this resolution. However, my concerns are as follows:

1. Costs of sequencing are very high, and an important question that ensues is how much investment should be put into genome sequencing. This is especially critical in low-income countries where perhaps an investment of money into detection or healthcare would be (arguably) more important. Yet the message that comes out from this paper is that it is important to sequence at least 0.5% of SARS-CoV-2 sequences in any country. I personally feel that there is way too much investment in genome sequencing in most countries, and much thought is required on WHY and WHEN sequencing is at all necessary (for example, many countries resorted to cheap PCR-based detection that may be much more worthwhile). This is of course a very tricky question but the way the paper is pitched now, I feel this side is completely lacking from the discussion.

2. The authors use simulations to explore when a variant can be detected. Would be nice to relax some of the assumptions - such as non-random sampling, which I am sure happens - and see what comes out.

3. Minor comments - In figure 1 I did not know what the country abbreviations stood for. Worth adding to legend.

Reviewer #3 (Remarks to the Author):

Brito et al. investigates the spatiotemporal heterogeneity in the global SARS-CoV-2 genomic surveillance and found that many LMIC sequenced less <0.5% of their cases.

This is nice work and the results are not surprising—undoubtedly, LMIC countries need further support on this front. The authors conclude that strengthening pathogen genomic surveillance efforts worldwide, but particularly in LMICs, should be a global priority to improve pandemic preparedness and suggest a benchmark for SARS-CoV-2 genomic surveillance efforts. I appreciate the benefits of genomic epidemiology and the use of sequence data to respond to this and other outbreaks has been instrumental in providing guidance. However, while the author's general advice and benchmark is well-intentioned, I am not entirely convinced that the narrative around it would benefit LMIC (if the expectation is for them to invest more on sequencing!) and I would like the author to further elaborate on this.

There are health response priorities such as massive testing, acquiring vaccines or addressing the

diversion of regular healthcare that could be more important during a global health emergency response in a LMIC than an accurate idea of what's circulating. The costs of implementing genomic epidemiology in LMIC may be higher than those of HIC (due to exchange rates, importation taxes, etc) so the authors should highlight the added value of it versus cheaper molecular testing. I think it is crucial to answer the question of what was done with the genomic surveillance of SARS-CoV-2 in LMIC that would have not been done without (other than triggering international travel restrictions)?

Conducting genomic surveillance activities, even in a non-emergency setting, already poses challenges to LMICs so it is unclear the benefit of such a big endeavour in the long-term (e.g. due to lack of secured funding). The benefits that genomic surveillance can bring to vaccine development are clearly not shared with these countries. So I believe the authors need to emphasise under which circumstances the use of sequencing can, realistically and quantitatively, influence the ability to respond in LMIC.

The authors make emphasis on the time between detecting a case and sequencing/reporting but how does this translate into improved response? For instance, Line 211 informs "This implies that limited genome sequencing intensity may affect the identification and response to new viral lineages with altered epidemiological and antigenic characteristics". In general, it can take a couple of months before a variant can be pointed as the culprit of an epidemiological change and it takes considerable knowledge transfer for LMIC countries to be able to do so on their own. I believe more focus is needed in what is directly actionable in LMIC from genomics rather than the opportunity of being able to meet a benchmark. While I appreciate the commitment of some (for instance, South Africa), I would be wary of suggesting similar efforts to other LMIC countries (+ a benchmark) without an idea of clear returns. Such an expensive endeavour can influence the ability to respond from those countries in other health areas and, in reality, could end-up just benefiting parachute science.

Minor remarks

Line 131 I believe the epidemic is also driven by the unequal access to vaccines (+ lack of knowledge and mistrust towards vaccines)

Line 138 "To help guide public health responses to evolving variants, it is essential to track the diversity of SARS-CoV-2 lineages circulating worldwide in near real-time"

Again, how exactly has this changed (and positively impacted) local policies?

Figure 1-Some countries are named-not clear why (mentioned in the text?)-but not consistently across panels. There is an issue with the merged pdf (labels, tick values and some legend data are not properly shown for figures).

Line 157 It would be nice to have some summary statistics (median/IQ) of the total number of confirmed cases of those countries to provide context

Line 160 I found this line hard to read

Line 183 Please provide details of what considered low COVID-19 incidence in this study

Line 183 What fraction of countries, from both regions, did not reach such a level? Same for Latin American countries (line 186).

I think that key findings, such as the fraction of LMICs vs HIC that sequenced 0.5% of cases or the fractions that had 'adequate' turnaround time, should be mentioned earlier.

Line 187 More details are needed here and it is not clear if this statement comes from a sensitivity analysis.

Line 191. I think this most 'general observation' should be mentioned before describing findings by income/region.

Line 197. It is not clear to me if the authors are suggesting that turnaround time was a response to the emergence of Alpha. But if so, I am not sure that I entirely agree. It may be the case that

the emergence of alpha coincided with countries catching up infrastructure development, supplies deployment, etc.

Line 230. I'm unsure which figure/result this conclusion comes from (fig 2A is capped at 5% prevalence) so this observation seems to be a very rare result.

Line 255 A probability of 0.2, as the probability of an event is a number between 0 and 1.

Line 310 UMC is not defined in the text

REVIEWER COMMENTS

Reviewer #1 (Remarks to the Author)

Brito et al present a formal analysis of the disparities in SARS-CoV-2 sequencing capacity. This is an interesting and important analysis that presents the current disparity in global SARS-CoV-2 sequencing.

I think the analysis are sound and the paper will be of great use to the scientific, and public health communities and will provide policy makers with useful data.

A major comment is that the authors analyse the problem and provide clear evidence of the inequalities in sequencing and suggest that “We found that sequencing at least 0.5% of the cases, with a TAT <21 days, could be a benchmark for SARS-CoV-2 genomic surveillance efforts.” But that is the limit of the recommendations for the future. I would suggest that this paper is an opportunity to present a vision of what a basic sequencing capacity every country should be aiming to have, with details about TAT, geographical coverage, data release, sample types, etc. A figure / flow chart summarising this would be useful.

We appreciate the reviewer’s point about providing more sequencing guidance to countries, but we believe that (i) genomic surveillance strategies should be context- and question-specific and that (ii) the currently cited WHO and ECDC documents already provide detailed guidance on a comprehensive range of topics related to SARS-CoV-2 genomic surveillance. These guideline documents also include documentation on sample size and sample selection, and discuss other aspects of targeted and representative sampling strategies, which are not the focus of our study. At the end of our manuscript, we now direct the readers towards these guidelines, and call for future research on the topic:

— **“Alongside with the guidance provided by the WHO and other public health authorities (see 14,15,38–43), ongoing surveys to understand barriers to virus genome sequencing and sampling selection strategies will provide valuable information for future surveillance programmes.”**

I have outlined some comments below that I think would improve the paper.

Query 1. Line 134 to 135 – there is no mention of B.1.1.529/Omicron?

Our initial analyses were performed prior to the emergence of B.1.1.529/Omicron. However, we have now included data submitted up to March 18th, 2022, in our analyses to cover the first two years of COVID-19 pandemic. We have made the following modification to the Methods section to reflect this:

— **“we used metadata related to the “country of exposure” of genomes submitted to GISAID up to March 18th, 2022, collected from EW (epidemiological week) 10 of 2020 (March 1st, 2020) to EW 8 of 2022 (February 26th, 2022).”**

Query 2. Line 142 – the last date seems a bit out of date? (I presume this reflects an initial submission a while ago). Could the authors update the numbers to be more contemporary?

Yes, the dataset used in the revised version of the manuscript now includes more recent data, as requested.

Query 3. Figure 1 – (A) There is no key to the colours in Figure 1? Other than the countries highlighted?

Thank you for this suggestion. A key is now provided in Figure 1, and the same colour scheme is replicated throughout the manuscript, with keys and legends where appropriate.

Query 4. Figure 1 panel. Could Figure 1 be split into two separate figures? I found it hard to extract information from the figures as the panels were so small. It is a pity because they are such great figures - it would be better if the reader can view them properly. Particularly figure 1A – readers need to see the details. Perhaps this could be Figure 1 and the rest of the panel (B, C, D) would move into a separate figure 2?

Yes, panels B, C, and D are now in a separate figure (Fig. 2). A spreadsheet with the results shown in Figure 1 is provided as Supplementary Material (Table S1).

Query 5. Line 188 to 191: I found this sentence particularly striking (and shocking). I would suggest that the authors include this point in the Abstract - as it would seem to be a clear objective that could, and should be solved in the immediate future.

After analysing the most recent data up to March 18th, 2022, we were pleased to observe that this finding no longer holds since countries that had no sequencing capacity at the time of our first submission have now been able to submit a limited amount of sequence data. However, sequencing intensities in LICs have not improved much. We now mention in the manuscript two important observations, highlighting unequal capacities of LMCs as compared to HIC and UMC:

— “By comparing the first and second years, however, important increases in surveillance intensity were observed in HICs, UMCs and LMCs, which expanded their weekly percentage of sequenced cases by 4.7, 15 and 22.5-fold, respectively. For LICs, no major improvements in surveillance intensity were observed.”

About timeliness, we mention that:

— “When we compare the TAT of countries based on their income classes, improvements were observed in all classes, except among low income countries, which had a higher median TAT in the second pandemic year (TAT median change from 71 to 109 days).”

Query 6. Line 225 – 227: The authors state here that these figures are independent of outbreak and population size. Did the authors attempt to include considerations of the physical size of the country, population densities and degree of interconnectivity between locations? For example – if a country was to heavily bias sampling in a capital city but then not sample in a secondary cities or locations (that might be many of thousands of kilometres away) and/or be more connected (in terms of transport) to a second country? It may well be that this kind of modelling is beyond the scope of this analysis, but at the least the authors should consider including some discussion around ensuring random sampling attempts to provide broad geographical coverage throughout a country rather than just focusing on one or two locations may risk an inaccurate picture of the true epidemiology. Of course, with the caveat that in resource limited situations something is better than nothing.

We thank the reviewer for this suggestion. We now include a novel supplementary figure (Fig. S9) with results of an extra analysis to address the question raised by the reviewer. Specifically, we looked at Denmark-wide lineage detection probabilities before they reach 100 cases from simulated sequencing results related to the capital region of Denmark alone. We feel like this is a fair compromise since heterogeneity in population density and mobility between countries make the parameter space for this type of analysis vast and potentially difficult for countries to identify themselves in this space.

We now mention in the revised manuscript:

— “By simulating a scenario of non-random sampling, focused in the most populous region of a country, we observed that the power to detect lineages decreases, but remains moderately useful when turnaround time is below 21 days, and sequencing intensity is at least 0.5% of all cases (Fig. S9).”

Query 7. Line 243 to 251: An additional analysis that wasn't conducted by the authors but would have added to this analysis - is sample selection. For example, in many locations sequencing capacity is often linked to healthcare in which patients often take a number of days or even weeks before being hospitalised with COVID-19 (and other infections). The usefulness and timeliness of sequencing to detect an emerging variant would likely be affected if samples were always taken from hospitalised patients - rather than catching the leading edge of the epidemic by sampling the most recent community samples. Some discussion along these lines should be included.

We agree with the Reviewer that sample selection is an important aspect of genomic epidemiology programs, but in our opinion the parameter space here is too vast to account for all potential biases that might exist in every country, and such analysis is outside the scope of this study.

Query 8. Line 326 - 338:

(a) The authors didn't include any issue around metadata and data deposition in public databases (e.g. the risk of release of patient identifiable data – particularly from populations who are more sensitive to data release and risks of deductive disclosure in remote communities) and then subsequent access to data. This has caused a major delay in the public release of data in a number of countries. Perhaps some comment along the lines that it is more important that data is deposited with a minimal data set (sampling date and country) rather than chasing perfection (e.g. more granular patient and geographical data).

(b) Also, restrictions on data access to data in GISAID have also caused problems. While I don't expect authors to specifically comment on GISAID directly (I note GISAID is included as an author) - a broader discussion of the importance of data sharing (timely and open access with respect to authorship of data generators) would be worthwhile – during the COVID-19 pandemic things are much better – there is still room for improvement.

We thank the Reviewer for this suggestion. We now address these points in the conclusion, and refer to the recent bulletin by the WHO, where these aspects are presented:

— “Adoption of **standardized protocols for representative genomic surveillance strategies** ^{16,36}, **establishment of data and minimal metadata standards, efficient and facilitated access to information, following equitable data sharing agreements (WHO 2022)**, and collaboration between academia, public health laboratories and other stakeholders will be essential to maximize cost-effectiveness and public health impact of genomic surveillance.”

Query 9. Line 345 to 347 – following on from point 6 – some discussion here about ensuring broad geographical coverage should be included.

Our study does not focus on evaluating or providing specific guidance on sampling strategies. However, as we pointed out in our reply above to Query 6, we now include an additional analysis looking at the impact of non-random sampling (see Fig. S9). We also include the following sentence in the conclusion of the manuscript:

— “While a random sampling strategy may provide accurate information about SARS-CoV-2 variant emergence and frequency estimation, we note that **genome sampling strategies should be considered pathogen- and question-specific.**”

This sentence is followed by specific references from international public health authorities, which provide more details about sampling strategies and other aspects of genomic surveillance:

- ECDC. *Sequencing of SARS-CoV-2 - first update* (2021).
- ECDC. *Detection and characterisation capability and capacity for SARS-CoV-2 variants within the EU/EEA* (2021).
- WHO. Global genomic surveillance strategy for pathogens with pandemic and epidemic potential 2022–2032. *Bulletin of the World Health Organization* vol. 100 239–239A (2022).
- ECDC. *Guidance for representative and targeted genomic SARS-CoV-2 monitoring* (2021).
- WHO. Genomic sequencing of SARS-CoV-2: a guide to implementation for maximum impact on public health (2021).
- WHO. *SARS-CoV-2 genomic sequencing for public health goals* (2021).

Query 10. Line 357 – 358: A bit more detail should be added here about other non-random sample selections including: (a) treatment failures (antivirals), (b) vaccine breakthrough, (c) reinfections, (d) clinical trials, (e) monitoring of immunocompromised infections, etc.

As detailed above in our reply to Query 9, we now cite several international guidelines of genomic surveillance, which go into more detail about the different uses of genomic sequencing, and include many examples of non-random sampling strategies.

Reviewer #2 (Remarks to the Author)

The paper of Brito and colleagues surveys and analyses the intensity of genomic surveillance for SARS-CoV-2 across the globe. They correlate between sequencing incidence and various features of a countries (socioeconomic factors and COVID-19 incidence). They further simulate how many sequences are needed to detect a given variant present at different frequencies. All in all, this is a very interesting and highly relevant topic that has never been explored before at this resolution. However, my concerns are as follows:

Query 1. Costs of sequencing are very high, and an important question that ensues is how much investment should be put into genome sequencing. This is especially critical in low-income countries where perhaps an investment of money into detection or healthcare would be (arguably) more important. Yet the message that comes out from this paper is that it is important to sequence at least 0.5% of SARS-CoV-2 sequences in any country. I personally feel that there is way too much investment in genome sequencing in most countries, and much thought is required on WHY and WHEN sequencing is at all necessary (for example, many countries resorted to cheap PCR-based detection that may be much more worthwhile). This is of course a very tricky question but the way the paper is pitched now, I feel this side is completely lacking from the discussion.

Thank you for raising these important points. We agree that cost-effective PCR-based tests can be extremely useful for SARS-CoV-2 surveillance after a VOC or VOI has been detected and characterised by full genome sequencing. We now explain the special cases where such types of tests are useful, and what they require:

— “In countries with limited sequencing capacity and/or long TATs, more **affordable PCR-based tests**, such as RT-PCR tests that distinguish VOCs based on target failures (for example, ‘S gene target failure’; SGTF), have been extremely valuable in providing evidence of the spread of a few variants, such as the VOCs Alpha and Ómicron, which contain specific deletions that lead to target failures. However, **VOC RT-PCR tests can only be deployed once local circulation of these lineages is confirmed using genome sequencing**. Without rapid sequencing and genomic characterization in the first place, as we observed for Omicron in late 2021, cost-effective PCR-based tests cannot be developed or deployed.”

For more details on the importance of sequencing, we now refer to several guidelines from international public health authorities (WHO, PAHO, ECDC) on genomic surveillance, including sampling strategy among other important technical aspects, which are not in the scope of the present study. We now reference these international guidelines in conclusion of our manuscript:

— “**Alongside the guidance provided by the WHO and other international public health authorities** (see 14,15,38–43), ongoing surveys to understand barriers to virus genome sequencing and sampling selection strategies will provide valuable information for future surveillance programmes.”

- ECDC. *Sequencing of SARS-CoV-2 - first update* (2021).
- ECDC. *Detection and characterisation capability and capacity for SARS-CoV-2 variants within the EU/EEA* (2021).
- WHO. Global genomic surveillance strategy for pathogens with pandemic and epidemic potential 2022–2032. *Bulletin of the World Health Organization* vol. 100 239–239A (2022).
- ECDC. *Guidance for representative and targeted genomic SARS-CoV-2 monitoring* (2021).
- WHO. *Genomic sequencing of SARS-CoV-2: a guide to implementation for maximum impact on public health* (2021).
- WHO. *SARS-CoV-2 genomic sequencing for public health goals* (2021).

Query 2. The authors use simulations to explore when a variant can be detected. Would be nice to relax some of the assumptions - such as non-random sampling, which I am sure happens - and see what comes out.

We have **included an additional set of supplementary figures (Figure S9)**, with results of a simulation (using data from Denmark) of a non-random sampling scenario where only the capital region of a country is sampled. We found that the power to detect lineages decreases when focusing only on the most populous region of the country but remains moderately useful with turnaround times <21 days and sequencing 0.5% of all cases.

Query 3. Minor comments - In figure 1 I did not know what the country abbreviations stood for. Worth adding to legend.

We now highlight with abbreviations only the countries with the highest overall percentage of sequenced cases in each region. The figure legend now includes the meaning of the ISO 3166-1 codes, as follows:

— “NZL = New Zealand; JPN = Japan; BRN = Brunei; MDV = Maldives; TJK = Tajikistan; ISR = Israel; DNK = Denmark; LUX = Luxembourg; POL = Poland; SVN = Slovenia; EGY = Egypt; GMB = Gambia; COG =

Republic of the Congo; DJI = Djibuti; BWA = Botswana; CAN = Canada; NIC = Nicaragua; BES = Bonaire; and SUR = Suriname.”

Reviewer #3 (Remarks to the Author)

Query 1. Brito et al. investigates the spatiotemporal heterogeneity in the global SARS-CoV-2 genomic surveillance and found that many LMIC sequenced less <0.5% of their cases. This is nice work and the results are not surprising—undoubtedly, LMIC countries need further support on this front. The authors conclude that strengthening pathogen genomic surveillance efforts worldwide, but particularly in LMICs, should be a global priority to improve pandemic preparedness and suggest a benchmark for SARS-CoV-2 genomic surveillance efforts. I appreciate the benefits of genomic epidemiology and the use of sequence data to respond to this and other outbreaks has been instrumental in providing guidance. However, while the author's general advice and benchmark is well-intentioned, I am not entirely convinced that the narrative around it would benefit LMIC (if the expectation is for them to invest more on sequencing!) and I would like the author to further elaborate on this.

We thank the reviewer for the positive assessment of our work, and agree that SARS-CoV-2 genomic data and analyses have been instrumental in providing guidance for public health responses. The large impact caused by VOCs worldwide, including in LMICs, highlights the importance of more equitable surveillance efforts. If governments in HICs and LMICs aim at acting based on evidence, receiving early warnings and understanding which locations need additional interventions, investments in surveillance are essential. However, as the reviewer pointed out, LMICs may have other, more urgent public health priorities, and would need some support to improve their local surveillance programmes. Hence, we modified the conclusion of the manuscript to stress this point:

— “Global efforts must be made to improve in-country genomic surveillance capacity, and to provide sustainable research funding for strengthening sequencing capacity and outbreak analytics, particularly in low and middle income countries.”

Query 2. There are health response priorities such as massive testing, acquiring vaccines or addressing the diversion of regular healthcare that could be more important during a global health emergency response in a LMIC than an accurate idea of what's circulating. The costs of implementing genomic epidemiology in LMIC may be higher than those of HIC (due to exchange rates, importation taxes, etc) so the authors should highlight the added value of it versus cheaper molecular testing. I think it is crucial to answer the question of what was done with the genomic surveillance of SARS-CoV-2 in LMIC that would have not been done without (other than triggering international travel restrictions)?

SARS-CoV-2 genome sequencing in LMICs has provided critical information on the early spread and transmissibility of novel variants of concern (Beta, Gamma, Delta, Omicron), but it also set the foundations for pandemic preparedness in areas that are most at risk for zoonotic emergence. Detection of novel VOCs would not have been possible with VOC-specific molecular testing, and routine pandemic preparedness relies on virus-specific and virus-agnostic genome sequencing strategies, as highlighted in the WHO's 2022–2032 global surveillance strategy cited in our study (WHO, Global genomic surveillance strategy for pathogens with pandemic and epidemic potential 2022–2032. *Bulletin of the World Health Organization* vol. 100 239–239A, 2022.

Beyond the closing sentence in the conclusion of the manuscript, we now also stress this aspect throughout the manuscript:

— “Future studies should focus on **additional variables that may affect genomic surveillance, especially in LMICs**, such as training laboratory and bioinformatics personnel, metadata standards, costs associated with imported consumables, and shipment delays that may be exacerbated by border closures and travel restrictions”

— “Strengthening virus genomic surveillance efforts worldwide, **but particularly in countries** where highly pathogenic viruses such as Ebola, yellow fever, monkeypox and polio are endemic, should be a global priority to improve pandemic preparedness.”

— “Global efforts must be made to improve in-country genomic surveillance capacity, and to **provide sustainable research funding for low and middle income countries...**”

As detailed in our Replies to Query 9 from Reviewer 1, and to Query 1 from Reviewer 2, we now refer to WHO and ECDC guidelines on SARS-CoV-2 genomic surveillance, that further highlight the importance of genomic sequencing in epidemiological surveillance.

Query 3. Conducting genomic surveillance activities, even in a non-emergency setting, already poses challenges to LMICs so it is unclear the benefit of such a big endeavour in the long-term (e.g. due to lack of secured funding). The benefits that genomic surveillance can bring to vaccine development are clearly not shared with these countries. So I believe the authors need to emphasise under which circumstances the use of sequencing can, realistically and quantitatively, influence the ability to respond in LMIC.

As the WHO states in its recent bulletin (“*Global genomic surveillance strategy for pathogens with pandemic and epidemic potential, 2022–2032*”), now cited in the conclusion of our manuscript, “Genomic surveillance is a powerful and proven tool that can help public health systems to detect, prepare for and respond to emerging pandemics and epidemics.” This is also mentioned in our response above to Query 2.

Variants with “greater transmissibility and potential immune escape” have been demanding more stringent policies, especially in locations where scientific evidence is available, and where evidence-based policies are valued and appropriately implemented. Policy makers cannot implement the necessary measures, proportionally to the risk posed by new pathogens, if they don’t know **‘what’** pathogen is present locally, **‘where’** it circulates in the community, **‘when’** that threat may arrive, **‘why’** it causes more risks, and **‘who’** are at more risk. Thanks to the timely work of researchers in South Africa, to mention a very recent example, the world knew **‘what’** Omicron represents, that it was spreading faster in many provinces of that country (**‘where’**), that other countries must be prepared as it could rapidly spread globally (**‘when’**), that it has greater transmissibility and immune escape (**‘why’**), and that it may pose greater risks, especially to individuals **‘who’** are more vulnerable (the elderly, immunocompromised) and/or not fully vaccinated with all the necessary booster doses. Without knowing ‘what’, ‘where’, ‘when’, ‘why’ and ‘who’, inefficient public health policies end up being implemented, and lives are unnecessarily impacted with high morbidity (long covid, sequelae) and high mortality.

This is a key question which we partly address in our response above to Query 2. In addition, we now include in the manuscript:

— “variants of concern (VOCs) such as Alpha/B.1.1.7; Beta/B.1.351; Gamma/P.1; Delta/B.1.617.2 and Omicron/B.1.1.529. These lineages pose increased global public health risks due to their greater transmissibility and potential immune escape from neutralizing antibodies induced by natural infections and/or vaccines. Variants of interest (VOIs) also require continued monitoring for changes in transmissibility, disease severity, or antigenicity.”

Query 4. The authors make emphasis on the time between detecting a case and sequencing/reporting but how does this translate into improved response? For instance, Line 211 informs “This implies that limited genome sequencing intensity may affect the identification and response to new viral lineages with altered epidemiological and antigenic characteristics”. In general, it can take a couple of months before a variant can be pointed as the culprit of an epidemiological change and it takes considerable knowledge transfer for LMIC countries to be able to do so on their own. I believe more focus is needed in what is directly actionable in LMIC from genomics rather than the opportunity of being able to meet a benchmark. While I appreciate the commitment of some (for instance, South Africa), I would be wary of suggesting similar efforts to other LMIC countries (+ a benchmark) without an idea of clear returns. Such an expensive endeavour can influence the ability to respond from those countries in other health areas and, in reality, could end-up just benefiting parachute science.

South Africa took an impressive 9 days to detect and provide evidence that Omicron poses a major threat to public health (collection date of first genome = November 13th, 2021; submission to open database = November 22nd, 2021). WHO classified Omicron as a variant of concern on November 26th, 2021 ([https://www.who.int/news/item/26-11-2021-classification-of-omicron-\(b.1.1.529\)-sars-cov-2-variant-of-concern](https://www.who.int/news/item/26-11-2021-classification-of-omicron-(b.1.1.529)-sars-cov-2-variant-of-concern)). This rapid detection and response, which took only 13 days and not a couple of months, was only possible because a robust genomic surveillance system was already in place in South Africa. The goal of our manuscript is not necessarily to encourage countries to sequence more SARS-CoV-2 genomes to reach a given benchmark, but to provide data-driven assessment of global surveillance efforts and provide considerations on genomic surveillance efforts for the timely detection of future disease outbreaks. There is no doubt that more equitable global genomic surveillance efforts would improve pandemic preparedness. Sustainable funding, global partnerships and fair data sharing protocols, including enforcement of the Nagoya protocol, should help to ensure LMICs can directly benefit locally from their improved genome sequencing capacity. We now highlight in the conclusion of the manuscript:

— “**global efforts** must be made to improve in-country genomic surveillance capacity and guarantee sustainable research funding for low and middle income countries”.

Minor remarks

Line 131 I believe the epidemic is also driven by the unequal access to vaccines (+ lack of knowledge and mistrust towards vaccines)

We agree with the reviewer. unequal access to vaccines has certainly played a huge role, particularly earlier in the pandemic. However, the closest known relatives of Omicron were sampled around early-to-mid 2020, when vaccines were not available at all, and thus likely would have emerged and spread even if the entire world had equal access to vaccines. We implemented the reviewer’s request in the manuscript, with the caveat stated here:

— More than two years into the COVID-19 pandemic, many countries continue to face large epidemics of SARS-CoV-2 infections (1), mostly driven by the emergence and spread of novel viral variants (2), **and unequal access to vaccines, particularly at the early stages of the pandemic..**

Line 138 “To help guide public health responses to evolving variants, it is essential to track the diversity of SARS-CoV-2 lineages circulating worldwide in near real-time”. Again, how exactly has this changed (and positively impacted) local policies?

Please see our responses above, where we explained how genomic epidemiology benefits public health responses. These and other points about the importance of genomic surveillance are also emphasized in distinct parts of our manuscript.

Figure 1-Some countries are named–not clear why (mentioned in the text?)–but not consistently across panels. There is an issue with the merged pdf (labels, tick values and some legend data are not properly shown for figures).

We now highlight with abbreviations only the countries with the highest overall percentage of sequenced cases in each region. The figure legend now includes the meaning of the ISO 3166-1 codes, as follows:

— “NZL = New Zealand; JPN = Japan; BRN = Brunei; MDV = Maldives; TJK = Tajikistan; ISR = Israel; DNK = Denmark; LUX = Luxembourg; POL = Poland; SVN = Slovenia; EGY = Egypt; GMB = Gambia; COG = Republic of the Congo; DJI = Djibuti; BWA = Botswana; CAN = Canada; NIC = Nicaragua; BES = Bonaire; and SUR = Suriname.”

Line 157 It would be nice to have some summary statistics (median/IQ) of the total number of confirmed cases of those countries to provide context

We now provide a new supplementary table with summary statistics related to sequencing and epidemiology in all countries included in the analyses (Table S3).

Line 160 I found this line hard to read

Thank you for pointing this out. We have updated and modified that sentence to read as follows:

— “Until late February 2022, while the total number of reported cases was relatively similar in high-income countries (HICs) and low/middle-income countries (LMICs) (i.e. 232.7 and 199.1 million cases, respectively), HICs shared 10-fold more sequences per COVID-19 case (3.53% and 0.35% sequenced cases, respectively) (Table S3).”

Line 183 Please provide details of what considered low COVID-19 incidence in this study

We now provide more details about the incidence levels considered in this manuscript, which correspond to US CDC guidelines in place up to late 2021: low incidence (<10 weekly cases per 100,000 people); moderate/substantial (10-99); high (>100 weekly cases per 100,000 people). We now refer to incidence levels accordingly:

— “Countries that faced mostly **moderate or lower incidences (below 100 cases per 100,000 people)** were more able to sequence higher proportions of cases (Fig. 1B; Fig. S3 and S4).”

— “Exceptionally, some countries, such as Denmark, Japan and the UK, despite facing scenarios of **high weekly COVID-19 incidence (>100 cases per 100,000 people)** in the first two years of the pandemic...”

— “Throughout the pandemic, many countries reported weekly incidences as **high as 100 cases per 100,000 inhabitants** (Fig. 1C, Fig. S3 and S4).”

Line 183 What fraction of countries, from both regions, did not reach such a level? Same for Latin American countries (line 186). I think that key findings, such as the fraction of LMICs vs HIC that sequenced 0.5% of cases or the fractions that had 'adequate' turnaround time, should be mentioned earlier.

Over the past months, the proportion of sequenced cases across the world improved in terms of sequencing intensity and timeliness. As a result of this improvement, we have now removed that sentence. A total of 58% (72 out of 124) of low and middle income countries have not reached the 0.5% threshold. Among high income countries, only 21.5% (14 out of 65) had less than 0.5% of their cases sequenced.

Line 187 More details are needed here and it is not clear if this statement comes from a sensitivity analysis.

We have removed this sentence and modified the paragraph.

Line 191. I think this most 'general observation' should be mentioned before describing findings by income/region.

The new version of the manuscript focuses on surveillance performance across countries (by income class). We point out the correlation with socio-economic factors as a potential explanation of the disparities, and mention that unlike in HICs, LMICs, no major improvements in sequencing intensity were observed in LICs.

Line 197. It is not clear to me if the authors are suggesting that turnaround time was a response to the emergence of Alpha. But if so, I am not sure that I entirely agree. It may be the case that the emergence of alpha coincided with countries catching up infrastructure development, supplies deployment, etc.

At the time of discovery, Alpha made a clear impact on the way public health organizations viewed genomic sequencing. For example, the ECDC's publications around the beginning of 2021 clearly indicate a preoccupation with variants of concern, beginning with publishing guidelines for sequencing on January 18th, 2021, to surveying domestic sequencing capacity across member states on February 16th, 2021, to announcing a programme where countries lacking sequencing capacity can send samples to ECDC's reference labs on June 16th, 2021. In all publications, the detection of variants of concern (mainly Alpha at the time) is directly named as a main objective. In the authors' personal experiences, the existence and detection of variants of concern fundamentally changed the way governments saw sequencing and we therefore disagree that turnaround time improvements just happened to coincide with the discovery of a SARS-CoV-2 lineage that was 70% more transmissible, an unprecedented shift in viral phenotype at the time.

We have updated that paragraph, and cited the references from the ECDC:

— "We observed that following the detection of more transmissible variants (VOCs) in late 2020, almost all geographic regions decreased their TAT, which allowed faster responses (Fig. 2C; see Fig. S6). Countries in Northern Europe, which have the fastest responses, decreased their median TAT from 20 to 10 days in the second pandemic year. **This change coincides with a series of bulletins and guidelines for SARS-CoV-2 sequencing, which were published by the ECDC in early 2021, specifically referring to the Alpha VOC 14–16.**"

Line 230. I'm unsure which figure/result this conclusion comes from (fig 2A is capped at 5% prevalence) so this observation seems to be a very rare result.

This figure comes from results presented in Table 1. We now make it clearer by citing the table earlier in that sentence.

— “However, under a scenario of random sampling, low income countries that sequence an average of 10 genomes per week may miss a SARS-CoV-2 lineage circulating at up to 21.7% prevalence (**Table 1**).”

Line 255 A probability of 0.2, as the probability of an event is a number between 0 and 1.

This is now included in the manuscript:

— “we found a 0.2 probability of detecting a lineage before it reaches 100 cases.

Line 310 UMC is not defined in the text

Thank you for pointing this out. A mention to this acronym is now included in distinct parts of the main text, and in Table 1:

— “and **upper/lower middle income countries (UMCs and LMCs)** had less than 0.5% of their cases sequenced in the first two years of pandemic”

REVIEWERS' COMMENTS

Reviewer #1 (Remarks to the Author):

No further comments. The authors have addressed all my comments satisfactorily.

Reviewer #4 (Remarks to the Author):

Brito & Semenova et al provide a noteworthy analysis of patterns of disparities in SARS-CoV-2 sequence generation efforts. They have updated the manuscript from original submission to include the Omicron (and sub lineage) variant and conducted a new analysis exploring the impact of non-random sampling on detection rates. Taken together, by analysing the contribution of labs to GISAID they identify trends in sequencing and make a suggestion that sequencing around 0.5% of cases with a turn-around-time of <21 days is a useful benchmark for CoV2 genomic surveillance.

I was specifically requested to look at the authors response to reviewers two and three, in particular focusing on the appropriate conveyance of the challenges around genomic surveillance implementation in LMICs. The manuscript explicitly compares HICs, UMCs and LMCs in its analysis justifying this as an important component.

It is of note that a thorough analysis of the factors which curtail the ambition for genomic surveillance globally is really beyond the scope of the paper. This is a discussion which necessitates coordinated infrastructure, policy and access to well documented socio-economic variables for joint economic/statistical assessment.

On the whole I am happy that the additional comments and citations the authors provide adequately accounts for the challenge of making sequencing recommendations, in particular for LMICs.

While the statement on '0.5% of total confirmed cases, with a TAT below 21 days' is followed by a comment on the need for public health implementation to support LMICs in the Discussion, I think a statement to this effect is needed in the abstract too to increase the palatability of the recommendation for public health agencies for which this is an overly ambitious target.

With this amendment I would be pleased to see the paper published and hope it encourages more investment in supporting sequencing capacity for SARS-CoV-2 and other notable pathogens.

REVIEWERS' COMMENTS

Reviewer #1 (Remarks to the Author):

No further comments. The authors have addressed all my comments satisfactorily.

Reviewer #4 (Remarks to the Author):

Brito & Semenova et al provide a noteworthy analysis of patterns of disparities in SARS-CoV-2 sequence generation efforts. They have updated the manuscript from original submission to include the Omicron (and sub lineage) variant and conducted a new analysis exploring the impact of non-random sampling on detection rates. Taken together, by analysing the contribution of labs to GISAID they identify trends in sequencing and make a suggestion that sequencing around 0.5% of cases with a turn-around-time of <21 days is a useful benchmark for CoV2 genomic surveillance.

I was specifically requested to look at the authors response to reviewers two and three, in particular focusing on the appropriate conveyance of the challenges around genomic surveillance implementation in LMCs. The manuscript explicitly compares HICs, UMCs and LMCs in its analysis justifying this as an important component.

It is of note that a thorough analysis of the factors which curtail the ambition for genomic surveillance globally is really beyond the scope of the paper. This is a discussion which necessitates coordinated infrastructure, policy and access to well documented socio-economic variables for joint economic/statistical assessment.

On the whole I am happy that the additional comments and citations the authors provide adequately accounts for the challenge of making sequencing recommendations, in particular for LMICs.

While the statement on '0/5% of total confirmed cases, with a TAT below 21 days' is followed by a comment on the need for public health implementation to support LMICs in the Discussion, I think a statement to this effect is needed in the abstract too to increase the palatability of the recommendation for public health agencies for which this is an overly ambitious target.

With this amendment I would be pleased to see the paper published and hope it encourages more investment in supporting sequencing capacity for SARS-CoV-2 and other notable pathogens.

We thank the reviewer for this suggestion. We have added a concluding sentence in the abstract, highlighting that global efforts must be done to provide financial support to LMICs, aiming at enabling a truly global surveillance network.

— “We found that sequencing around 0.5% of the cases, with a TAT <21 days, could provide a benchmark for SARS-CoV-2 genomic surveillance. Socioeconomic inequalities undermine the global pandemic preparedness, and **efforts must be made to support LICs and MICs improve their local sequencing capacity.**”